# Test-Time Alignment of LLMs via Sampling-Based Optimal Control in pre-logit space

## Abstract

Test-time alignment of large language models (LLMs) attracts attention because fine-tuning LLMs requires high computational costs. In this paper, we propose a new test-time alignment method called adaptive importance sampling on pre-logits (AISP) on the basis of the sampling-based model predictive control with the stochastic control input. AISP applies the Gaussian perturbation into pre-logits, which are outputs of the penultimate layer, so as to maximize expected rewards with respect to the mean of the perturbation. We demonstrate that the optimal mean is obtained by importance sampling with sampled rewards. AISP outperforms best-of-n sampling in terms of rewards over the number of used samples and achieves higher rewards than other reward-based test-time alignment methods.

## 1 Introduction

Alignment of large language models (LLMs) is a vital technique to enable the safe and widespread use of LLMs in real-world applications. A promising alignment method is reinforcement learning from human feedback (RLHF) (Ouyang et al., 2022; Christiano et al., 2017; Ziegler et al., 2019; Bai et al., 2022). However, RLHF imposes a heavy computational burden since fine-tuning LLMs requires high computational costs (Rafailov et al., 2023; Kong et al., 2024; Hu et al., 2022). To address this, test-time (also known as inference-time and decoding-time) alignment attracts attention (Kong et al., 2024; Li et al., 2024a; Snell et al., 2024; Huang et al., 2025; Li et al., 2024b).

Test-time alignment aligns LLMs with human preference without updating parameters of LLMs. This paper focuses on test-time alignment methods that find the optimal responses in terms of maximizing the score of a given reward model. To this goal, best-of-n sampling (BoN) is a simple but effective method, which selects the response that achieves the highest reward values from $N$ generated responses from the base LLMs (Snell et al., 2024; Lightman et al., 2023; Brown et al., 2024; Sessa et al., 2025). Though BoN can asymptotically optimize the same objective function as KL-constrained reinforcement learning (RL) (Yang et al., 2024), there might be room for improvements, such as in sample efficiency, because it does not actively explore the optimal responses. As another line of research, Kong et al. (2024) formalized test-time alignment as the optimal control problem and proposed RE-Control inspired by control theory. RE-Control applies an external control signal to the representations of LLMs and optimizes these input trajectories. Though RE-Control can actively explore the optimal responses by the control input, it needs to train a value function using a reward model: i.e., it requires computation and storage costs for training including dataset collection. *Can LLMs be controlled by the training-free methods to explore the optimal response?*

In this paper, we propose a new test-time alignment inspired by sampling-based control methods without a training process. Traditional optimal control theory can optimize input trajectories without any training process by solving differential equations such as the Pontryagin's maximum principle. However, these methods are not applicable to LLM alignment because LLMs are nonlinear, complicated and large-scale systems (Chen et al., 2024). For such systems, sampling-based model predictive control has been advanced by leveraging the parallel computing capabilities of GPUs (Williams et al., 2018; 2017). Therefore, we adopt this optimal control technique in the LLM alignment by incorporating adaptive importance sampling. First, we formalize LLM alignment as a stochastic control problem where the control input is a stochastic perturbation on pre-logits, which are outputs of the penultimate layer of the LLM. In our formulation, the perturbation follows a Gaussian distribution. According to this, the distribution of pre-logit sequences also becomes a Gaussian dis-

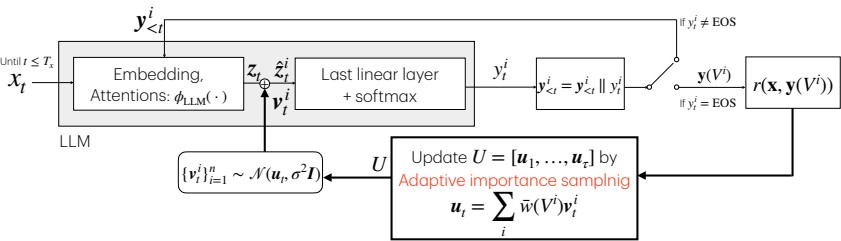

Figure 1: Illustration of AISP. $n$ input trajectries $\{\{\boldsymbol{v}_t^i\}_{t=1}^{\mathcal{T}}\}_{i=1}^n$ are sampled from $\mathcal{N}(\boldsymbol{u}_t, \sigma^2\boldsymbol{I})$. The input $\boldsymbol{v}_t^i$ is added to the pre-logit $\boldsymbol{z}_t$, which is obtained by applying LLMs to the past tokens $\boldsymbol{y}_{<t}^i$. The $t$-th token $y_t^i$ is sampled and concatenated with the past tokens $\boldsymbol{y}_{<t}^i$. When $y_t^i$ is the end-of-sequence token, the rewards of $\{\mathbf{y}(V^i)\}_{i=1}^n$ are evaluated and used in adaptive importance sampling for $\boldsymbol{u}_t$.

tribution and can be written by the closed form, unlike token sequence distributions. Next, we derive the optimal distribution through the free energy that bounds this problem. Since this distribution is intractable, we approximate it by using importance sampling where the weighting function can be easily computed thanks to the Gaussian assumption. We iteratively update the proposal distribution via adaptive importance sampling (Kloek & Van Dijk, 1978; Cappé et al., 2004; Bugallo et al., 2017) because naïve importance sampling can require a large number of effective samples due to the vast pre-logit sequence space. Therefore, our method is called adaptive importance sampling on pre-logits (AISP, Fig. 1). After explanation of AISP, we discuss the connection between the Gaussian assumption and the last softmax layer in neural networks. Additionally, we reveal that AISP becomes equivalent to BoN with the specific sampling strategy in the limit of a hyperparameter. Experiments demonstrate that AISP increases reward values faster than BoN in terms of the number of used generated samples. Additionally, AISP also outperforms RE-Control even though it does not require training dataset collection in advance. Since AISP requires fewer samples than BoN, we also evaluate Batched AISP, which simultaneously handles multiple prompts with small samples, and confirm that Batched AISP can outperform BoN under the same iterations.

## 2 PRELIMINARY

### 2.1 BEST-OF-N SAMPLING

Let $x_t, y_t \in \mathcal{V}$ denote tokens in a vocabulary space $\mathcal{V}$ at the $t$-th position. Given an input prompt $\mathbf{x} = [x_1, \ldots, x_{T_x}]$, an LLM generates a response $\mathbf{y} = [y_1, \ldots, y_{T_y}]$ from the probability $P_{\text{LLM}}(\cdot|\mathbf{x})$. Best-of-N sampling (BoN) attempts to generate aligned responses based on a given reward model $r(\mathbf{x}, \mathbf{y}) \in \mathbb{R}$. BoN samples $N$ responses from the base LLM as $\mathbf{y} \sim P_{\text{LLM}}(\cdot|\mathbf{x})$ and constructs a set $\mathcal{Y}_N = [\mathbf{y}^1, \ldots, \mathbf{y}^N]$. Next, BoN selects the best sample from the set $\mathcal{Y}_N$ as

$$\mathbf{y}_{\text{BoN}} = \arg\max_{\mathbf{y} \in \mathcal{Y}_N} r(\mathbf{x}, \mathbf{y}). \tag{1}$$

This simple algorithm is an effective and popular method to align LLMs (Lightman et al., 2023; Snell et al., 2024; Sessa et al., 2025). Yang et al. (2024) have shown that BoN asymptotically optimizes the following objective function of KL-constrained RL:

$$\max_{\pi(\cdot|\mathbf{x})} \mathbb{E}_{\mathbf{y} \sim \pi(\cdot|\mathbf{x})} r(\mathbf{x}, \mathbf{y}) - \lambda D_{KL}(\pi(\cdot|\mathbf{x})|P_{\text{LLM}}(\cdot|\mathbf{x})). \tag{2}$$

$D_{KL}(\pi(\cdot|\mathbf{x})|P_{\text{LLM}}(\cdot|\mathbf{x}))$ prevents $\pi(\cdot|\mathbf{x})$ from moving far away from the base LLM $P_{\text{LLM}}(\cdot|\mathbf{x})$. Equation (2) has a closed solution (Beirami et al., 2024; Korbak et al., 2022; Go et al., 2023):

$$\pi^*(\mathbf{y}|\mathbf{x}) = \frac{1}{\eta} P_{\text{LLM}}(\mathbf{y}|\mathbf{x}) \exp(\frac{1}{\lambda} r(\mathbf{x}, \mathbf{y})), \tag{3}$$

where $\eta = \sum_{\mathbf{y}} P_{\text{LLM}}(\mathbf{y}|\mathbf{x}) \exp(\frac{1}{\lambda} r(\mathbf{x}, \mathbf{y}))$ is a normalization constant, which is hard to estimate (Rafailov et al., 2023).

### 2.2 RE-CONTROL

Kong et al. (2024) have formulated the LLM alignment as the optimal control problem where control input $\boldsymbol{u}_t \in \mathbb{R}^d$ is added to the representation of an auto-regressive LLM as

$$y_t \sim \text{softmax}(\boldsymbol{W}_{\text{LLM}}(\boldsymbol{z}_t + \boldsymbol{u}_t) + \boldsymbol{b}_{\text{LLM}}). \tag{4}$$

where $\boldsymbol{W}_{\text{LLM}}$ and $\boldsymbol{b}_{\text{LLM}}$ are the parameter of the last linear layer of the LLM. $\boldsymbol{z}_t \in \mathbb{R}^d$ is called *pre-logit*[1], which is the output vector of the penultimate layer of the LLM: $\boldsymbol{z}_t = \phi_{\text{LLM}}(\boldsymbol{y}_{<t})$ where $\boldsymbol{y}_{<t} = [y_0, \dots, y_{t-1}]$ are the past tokens including the input prompt x. $\phi_{\text{LLM}}(\cdot)$ contains an embedding layer and attention layers. In this formulation, $\boldsymbol{u}_t$ is optimized through the gradient ascent to maximize the value function $V(\boldsymbol{z}_t)$, which evaluates the current state of LLMs to maximize rewards at the terminal. However, this value function needs to be trained on the dataset, which are composed of various states, responses, and rewards: $D_V = \{(\boldsymbol{z}_{0:T}^i, \boldsymbol{y}^i, r(\mathbf{x}^i, \mathbf{y}^i))\}_{i=1}^M$. In fact, Kong et al. (2024) uses 349,000 training prompts in SHP (Ethayarajh et al., 2022) to collect them, which incurs storage costs and training time. Instead of training the value function, we adopt a sampling-based optimal control to LLM alignment.

## 3 PROPOSED METHOD: AISP

To maximize rewards, we consider applying the control theory to LLM alignment similar to Kong et al. (2024), but without training. In fact, traditional optimal control methods do not require any training process because the optimal input trajectories are derived by solving differential equations such as the Pontryagin's maximum principle. However, such methods are ineffective for LLM alignment because LLMs are nonlinear large-scale systems (Chen et al., 2024). For such systems, recent optimal control methods have incorporated a sampling-based approach with model predictive control by considering stochastic input. Thus, we adopt the stochastic optimal control method called model predictive path integral control (MPPI) (Williams et al., 2018; 2017) to LLM alignment. First, we formalize our problem and explain the closed solution. Since it is an intractable distribution, we present adaptive importance sampling to solve this problem. Next, we discuss the connection between the assumption in pre-logit and softmax function, and the connection with BoN. Finally, we explain the details of implementation.

### 3.1 PROBLEM FORMULATION

Whereas RE-Control (Eq. (4)) uses the deterministic input $\boldsymbol{u}_t$, we apply the stochastic control input $\boldsymbol{v}_t$ with mean $\boldsymbol{u}_t$ to LLMs and optimize $\boldsymbol{u}_t$. Specifically, we inject a Gaussian noise $\boldsymbol{v}_t \in \mathbb{R}^d$ to pre-logit $\boldsymbol{z}_t = \phi_{\text{LLM}}(\boldsymbol{y}_{<t})$ for the time interval $t \in [1, \tau]$. The input prompt x corresponds to $\boldsymbol{y}_{<1}$. Then, a pre-logit follows a Gaussian distribution, and the $t$-th token is given by

$$y_t = \begin{cases} \arg\max_i \left[\text{softmax}(\boldsymbol{W}_{\text{LLM}}(\boldsymbol{z}_t + \boldsymbol{v}_t) + \boldsymbol{b}_{\text{LLM}}), \boldsymbol{v}_t \sim \mathcal{N}(\boldsymbol{u}_t, \sigma^2 \boldsymbol{I})\right]_i, & \text{for } 1 \le t \le \tau, \\ \arg\max_i \left[\text{softmax}(\boldsymbol{W}_{\text{LLM}}\boldsymbol{z}_t + \boldsymbol{b}_{\text{LLM}})\right], & \text{for } \tau < t. \end{cases} \quad (5)$$

where $\sigma^2 \boldsymbol{I}$ is a covariance matrix with a fixed variance $\sigma^2 \in \mathbb{R}$. This can be interpreted as the distribution of the pre-logit $\hat{\boldsymbol{z}}_t$ in AISP is given by $p(\hat{\boldsymbol{z}}_t|\boldsymbol{y}_{<t}) = \mathcal{N}(\phi_{\text{LLM}}(\boldsymbol{y}_{<t}) + \boldsymbol{u}_t, \sigma^2 \boldsymbol{I})$ for $t \in [1, \tau]$. The distribution of the input trajectory $V = [\boldsymbol{v}_1, \dots, \boldsymbol{v}_\tau] \in \mathbb{R}^{d \times \tau}$ is a joint Gaussian distribution:

$$q(V|U, \sigma^2) = \frac{1}{(2\pi\sigma^2)^{\frac{d\tau}{2}}} \exp\left(-\frac{1}{2\sigma^2} \sum_{t=1}^{\tau} (\boldsymbol{v}_t - \boldsymbol{u}_t)^\top (\boldsymbol{v}_t - \boldsymbol{u}_t)\right), \quad (6)$$

where $U = [\boldsymbol{u}_1, \dots, \boldsymbol{u}_T] \in \mathbb{R}^{d \times \tau}$ is the mean of the input trajectory. Let $\mathbb{Q}_{U,\sigma^2}$ be the distribution corresponding to the density function $q(V|U, \sigma^2)$. Similar to the objective of KL-constrained RL Eq. (2), we optimize the expected reward values with the KL constraint as

$$\min_U J(\mathbf{x}, U) = \min_U -\mathbb{E}_{V \sim \mathbb{Q}_{U,\sigma^2}}\left[r(\mathbf{x}, \mathbf{y}(V))\right] + \lambda \mathbb{D}_{\text{KL}}(\mathbb{Q}_{U,\sigma^2}|\mathbb{P}) \quad (7)$$

where $\mathbf{y}(V) = [y_1, \dots, y_{T_y}]$ is a response generated by Eq. (5). $\lambda \mathbb{D}_{\text{KL}}(\mathbb{Q}|\mathbb{P})$ is the regularization term so that the resulting distribution does not deviate from the base LLM where $\lambda > 0$ is a hyper-parameter. To satisfy this, a base distribution $\mathbb{P}$ should be a zero-mean Gaussian distribution:

$$p(V|\boldsymbol{0}, \sigma^2) = \frac{1}{(2\pi\sigma^2)^{\frac{d\tau}{2}}} \exp\left(-\frac{1}{2\sigma^2} \sum_{t=1}^{\tau} \boldsymbol{z}_t^\top \boldsymbol{z}_t\right). \quad (8)$$

This implies that we assume the pre-logit distribution of the base LLM following $p_{\text{LLM}}(\boldsymbol{z}_t|\boldsymbol{y}_{<t}) = \mathcal{N}(\phi_{\text{LLM}}(\boldsymbol{y}_{<t}), \sigma^2 \boldsymbol{I})$. The KL-divergence $\mathbb{D}_{\text{KL}}(\mathbb{Q}_{U,\sigma^2}|\mathbb{P})$ is given by $\mathbb{D}_{\text{KL}}(\mathbb{Q}_{U,\sigma^2}|\mathbb{P}) = 1/2\sigma^2 \sum_{t=1}^{\tau} \boldsymbol{u}_t^\top \boldsymbol{u}_t$. After the optimization, we can obtain the optimal reward $\mathbf{y}_{\text{AISP}}$ as $\mathbf{y}_{\text{AISP}} = \mathbf{y}(V = U^*)$ where $U^*$ is the solution of Eq. (7). Since we can generate several candidate responses at the last, we select $\mathbf{y}_{\text{AISP}} = \arg\max_{V \in \mathcal{V}} \mathbf{y}(V)$ where $\mathcal{V} = \{V^i | V^i \sim q(V|U^*, \sigma^2)\}$ instead of just using $\mathbf{y}(V = U^*)$.

---

[1]Though Kong et al. (2024) called $\boldsymbol{z}_t$ logit, we call it pre-logit to distinguish it from the the input to the softmax function: $\boldsymbol{W}_{\text{LLM}}\boldsymbol{z}_t + \boldsymbol{b}_{\text{LLM}}$.

## 3.2 Free energy and Optimal distribution

Optimization problems such as Eq. (7) correspond to the optimal control problems called model predictive path integral control (MPPI) (Williams et al., 2018; 2017) and can be solved by considering a certain free energy. To optimize Eq. (7), we consider the following free energy:

$$F(r, p, \mathbf{x}, \lambda) = \log \left( \mathbb{E}_{V \sim \mathbb{P}} \left[ \exp \left( \tfrac{1}{\lambda} r(\mathbf{x}, \mathbf{y}(V)) \right) \right] \right). \tag{9}$$

By using $\mathbb{Q}$ and Jensen's inequality, we have the following result:

**Theorem 3.1.** *Free energy Eq. (9) satisfies* $-\lambda F(r, p, \mathbf{x}, \lambda) \leq J(\mathbf{x}, U)$ *and the equality holds if*

$$q^*(V) = \tfrac{1}{\eta} \exp \left( \tfrac{1}{\lambda} r(\mathbf{x}, \mathbf{y}(V)) \right) p(V) \tag{10}$$

*where $\eta$ is a normalization constant given by* $\eta = \int_{\mathbb{R}^{d \times \tau}} \exp \left( \tfrac{1}{\lambda} r(\mathbf{x}, \mathbf{y}(V)) \right) p(V) dV.$

All of the proofs can be found in A. This theorem shows that the free energy Eq. (9) is the lower bound of the objective function Eq. (7), and the optimal density function is given by Eq. (10). The optimal density function Eq. (10) is intractable, and it is difficult to obtain directly. Next, we show how to approximate it by using importance sampling. Note that while the similar result Eq. (3) has been presented for the distribution of responses, Eq. (10) is related to the distribution of pre-logits. This difference makes the optimization easy because $p(V) = p(V|\mathbf{0}, \sigma)$ in Eq. (10) is a tractable distribution, and its sampling is easy as written in the next subsection.

## 3.3 Adaptive importance sampling

We consider to approximate $\mathbb{Q}^*$ of Eq. (10) by the Gaussian distribution of Eq. (6) though importance sampling (Robert et al., 1999; Kloek & Van Dijk, 1978; Cappé et al., 2004; Bugallo et al., 2017). We recall the following theorem, which was established in Williams et al. (2018):

**Theorem 3.2.** *(Williams et al., 2018) The KL divergence* $\mathbb{D}_{\mathrm{KL}}(\mathbb{Q}^* | \mathbb{Q}_{U, \sigma^2})$ *is minimized by* $U^* = [\boldsymbol{u}_1^*, \ldots, \boldsymbol{u}_\tau^*]$ *where*

$$\boldsymbol{u}_t^* = \mathbb{E}_{V \sim \mathbb{Q}^*}[\boldsymbol{v}_t]. \tag{11}$$

*Let* $q(V|\hat{U}, \sigma^2)$ *and* $\mathbb{Q}_{\hat{U}, \sigma^2}$ *be a proposal density function for importance sampling and the corresponding distribution, respectively. Equation (11) is re-written as* $\mathbb{E}_{V \sim \mathbb{Q}^*}[\boldsymbol{v}_t] = \mathbb{E}_{V \sim \mathbb{Q}_{\hat{U}, \sigma^2}}[w(V)\boldsymbol{v}_t]$, *where* $w(V)$ *is the weight function given by*

$$w(V) = \tfrac{1}{\eta} \exp \left( \tfrac{1}{\lambda} r(\mathbf{x}, \mathbf{y}(V)) - \tfrac{1}{\sigma^2} \sum_{t=1}^{\tau} \hat{\boldsymbol{u}}_t^\top \boldsymbol{v}_t - \tfrac{1}{2} \hat{\boldsymbol{u}}_t^\top \hat{\boldsymbol{u}}_t \right). \tag{12}$$

This theorem indicates that Eq. (11) can be approximated by importance sampling where the weight function is Eq. (12). We generate $n$ samples $\{V^i\}_{i=1}^n$ from the proposal distribution $\mathbb{Q}_{\hat{U}, \sigma^2}$ and approximate $U^*$ as $\hat{\boldsymbol{u}}_t^* = \sum_i (w(V^i)/\sum_j w(V^j))\boldsymbol{v}_t^i = \sum_i \bar{w}^i \boldsymbol{v}_t^i$ where $\sum_j w(V^j)$ is empirical normalization instead of $\eta$. The $i$-th weight $\bar{w}^i$ is given by

$$\bar{w}^i = \frac{\exp \left( \tfrac{1}{\lambda} r(\mathbf{x}, \mathbf{y}(V^i)) - \tfrac{1}{\sigma^2} \sum_{t=1}^{\tau} \hat{\boldsymbol{u}}_t^\top \boldsymbol{v}_t^i \right)}{\sum_j \exp \left( \tfrac{1}{\lambda} r(\mathbf{x}, \mathbf{y}(V^j)) - \tfrac{1}{\sigma^2} \sum_{t=1}^{\tau} \hat{\boldsymbol{u}}_t^\top \boldsymbol{v}_t^j \right)}, \tag{13}$$

which is implemented by using a softmax function. In this computation, the term of $\hat{\boldsymbol{u}}_t^\top \hat{\boldsymbol{u}}_t$ in Eq. (12) is canceled between the numerator and denominator. Then, the optimal $U^*$ can be obtained through importance sampling. Since importance sampling requires a lot of samples if the proposal distribution $q(V|\hat{U}, \sigma^2)$ is far from $\mathbb{Q}^*$, we exploit the adaptive importance sampling (Cappé et al., 2004; Bugallo et al., 2017), which is iterative importance sampling. Specifically, we updates $\hat{\boldsymbol{u}}_t$ by using importance sampling with $n$ sample for $\kappa$ iterations:

$$\hat{\boldsymbol{u}}_t^{k+1} = \sum_{i=1}^n \bar{w}^i \boldsymbol{v}_t^{i,k}, \quad \boldsymbol{v}_t^{i,k} \sim \mathcal{N}(\hat{\boldsymbol{u}}_t^k, \sigma^2 \boldsymbol{I}) \tag{14}$$

for $k = 1, \ldots, \kappa$. Similar to BoN, we select the best sample from all samples in the computation of AISP: $\mathbf{y}_{\mathrm{AISP}} = \operatorname{argmax}_{i,k} \mathbf{y}(V^{i,k})$. The algorithm of AISP can be found in Appendix B.

Note that our formulation is followed by MPPI (Williams et al., 2018) and can actually be extended to MPPI on pre-logits by changing sampling methods. While MPPI moves prediction and control windows by determining and applying $\hat{\boldsymbol{u}}_1$ to the control target for each iteration, AISP uses a fixed control window $t \in [0, \tau]$. This is because once moving windows fix the prefix tokens, generated responses lose diversity. AISP can explore large response spaces by using the fixed window starting $t = 0$ and adaptive importance sampling.

## 3.4 MODELING PRE-LOGITS DISTRIBUTIONS BY GAUSSIAN DISTRIBUTIONS

As explained above, AISP assumes that the pre-logit distribution follows a Gaussian distribution as $p(\hat{z}_t|\boldsymbol{y}_{<t}) = \mathcal{N}(\phi_{\text{LLM}}(\boldsymbol{y}_{<t}) + \boldsymbol{u}_t, \sigma^2 \boldsymbol{I})$. In this section, we discuss the reason why this assumption reduces the difficulty in the optimization, and the connection between this assumption and the output softmax layer in neural language models.

**How the Gaussian assumption simplifies the problem** As described in Theorem 3.2, the Gaussian assumption derives a simple optimization procedure by using importance sampling. This is because the KL divergence between the optimal distribution and the Gaussian distribution is minimized by the expectation of $\boldsymbol{v}_t$ over $\mathbb{Q}^*$, and the weight function becomes simple because input trajectories also follow the Gaussian distribution. If we impose no constraints on the prior distribution, this computation requires modeling techniques for complicated distributions such as normalizing flows (Power & Berenson, 2023) and does not yield a simple test-time alignment.

**Connection with softmax output layer** The Gaussian assumption is related to the implicit assumption of pre-logits by the softmax layer. An auto-regressive LLM generally uses a softmax function with a linear layer as the last layer:

$$P_{\text{LLM}}(y_t = y^i|\boldsymbol{y}_{<t}) = \frac{\exp(\boldsymbol{w}_i^\top \boldsymbol{z}_t + \boldsymbol{b}_i)}{\sum_{j=1}^{|\mathcal{V}|} \exp(\boldsymbol{w}_j^\top \boldsymbol{z}_t + \boldsymbol{b}_j)}. \tag{15}$$

The softmax function is derived when the conditional distribution of pre-logits $p(\boldsymbol{z}_t|y_t = y^i)$ is an exponential family distribution.[2] If $p(\boldsymbol{z}_t|y_t = y^i)$ is a Gaussian distribution as $p(\boldsymbol{z}_t|y_t = y^i) = \mathcal{N}(\boldsymbol{\mu}_{y^i}, \boldsymbol{\Sigma})$, we have the following equality from Bayes' theorem:

$$P(y_t = y^i|\boldsymbol{z}_t) = \frac{p(\boldsymbol{z}_t|y_t=y^i)P(y_t=y^i)}{p(\boldsymbol{z}_t)} = \frac{p(\boldsymbol{z}_t|y_t=y^i)P(y_t=y^i)}{\sum_j^{|\mathcal{V}|} p(\boldsymbol{z}_t|y_t=y^j)P(y_t=y^j)} \tag{16}$$

$$= \frac{\exp(\boldsymbol{\mu}_{y^i}^T \boldsymbol{\Sigma}^{-1}\boldsymbol{z} - \frac{1}{2}\boldsymbol{\mu}_{y^i}\boldsymbol{\Sigma}^{-1}\boldsymbol{\mu}_{y^i} + \ln P(y_t=y^i))}{\sum_j^{|\mathcal{V}|} \exp(\boldsymbol{\mu}_{y^j}^T \boldsymbol{\Sigma}^{-1}\boldsymbol{z} - \frac{1}{2}\boldsymbol{\mu}_{y^j}\boldsymbol{\Sigma}^{-1}\boldsymbol{\mu}_{y^j} + \ln P(y_t=y^j))}. \tag{17}$$

This function corresponds to the softmax function with a linear layer such that $\boldsymbol{w}_i = \boldsymbol{\mu}_{y^i}^\top \boldsymbol{\Sigma}^{-1}$ and $\boldsymbol{b}_i = -\frac{1}{2}\boldsymbol{\mu}_{y^i}\boldsymbol{\Sigma}^{-1}\boldsymbol{\mu}_{y^i} + \ln P(y_t = y^i)$ . From this result, Lee et al. (2018) assume that the pre-trained neural classifier has the pre-logtis following the Gaussian distribution given a class label in image recognition. Since neural language models also use softmax and cross-entropy loss, we can hypothesize that the trained pre-logit distribution $p(\boldsymbol{z}_t|y_t)$ follows a Gaussian distribution. From this perspective, AISP can be regarded as exploring the optimal pre-logit distribution $p(\hat{\boldsymbol{z}}_t|y_t^*)$ under the assumption, where the distribution given the optimal response can be decomposed as $p(\hat{\boldsymbol{z}}_1, \ldots, \hat{\boldsymbol{z}}_t|\mathbf{y}^*) = \prod_t p(\hat{\boldsymbol{z}}_t|y_t^*)$.

## 3.5 CONNECTION WITH BON

In AISP, $\lambda$ is the temperature parameter in softmax of Eq. (13). Since softmax is a smoothed approximation of argmax and $\lambda > 0$, Eq. (13) is asymptotically close to argmin when $\lambda \to 0^+$. Therefore, we have the following result:

**Theorem 3.3.** *When $\lambda \to 0^+$ and $\kappa = 1$, AISP becomes BoN with the candidate set $\mathcal{Y}_n$ as*

$$\mathcal{Y}_n = \{\mathbf{y}(V^i)|V^i \sim q(V|\hat{U}, \sigma^2), i = 1, \ldots, n\}. \tag{18}$$

From this result, AISP can be regarded as a continuous approximation of BoN with a specific sampling strategy. In other words, AISP is a generalization of BoN, and AISP subsumes BoN.

## 3.6 IMPLEMENTATION

We have explained the formulation of AISP. In this section, we will explain the technique to enhance the practical performance and parallelism in implementation.

---

[2]This section considers a conditional distribution given an output token $y_t$ not given the past tokens $\boldsymbol{y}_{<t}$.

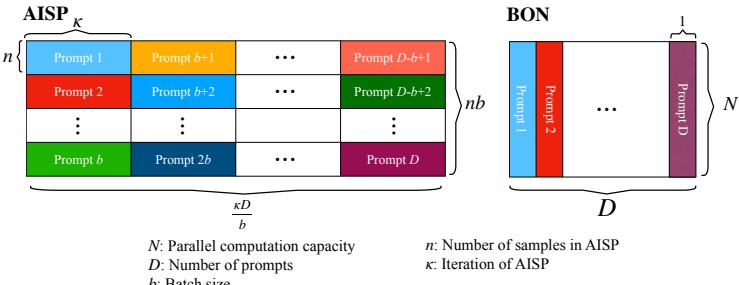

Figure 2: Schematic illustration of computational costs (vertical: parallelism; horizontal: iterations) of Batched AISP and BoN for $D$ prompts. When $\kappa D/b = D, nb = N \Leftrightarrow \kappa = b, n = N/b$, Batched AISP and BoN have almost the same sequential and parallel computational cost.

**Relaxation of constraints** As discussed above, $\lambda$ can be regarded as a temperature parameter. Small $\lambda$ allows deviation from the base LLM, but large $\lambda$ causes numerical instability (Williams et al., 2018). To achieve both numerical stability and large penalties, we relax $\mathbb{P}$ as $p(V|\alpha\hat{U}, \sigma)$ from $p(V|\mathbf{0}, \sigma)$ where $0 < \alpha < 1$ by introducing the technique in MPPI (Williams et al., 2018). Under this relaxation, the weight $\bar{w}^i$ becomes

$$\bar{w}^i = \frac{\exp\left(\frac{1}{\lambda}r(\mathbf{x}, \mathbf{y}(V^i)) - \frac{1-\alpha}{\sigma^2}\sum_{t=1}^{\tau}\hat{\boldsymbol{u}}_t^\top \boldsymbol{v}_t^i\right)}{\sum_j \exp\left(\frac{1}{\lambda}r(\mathbf{x}, \mathbf{y}(V^j)) - \frac{1-\alpha}{\sigma^2}\sum_{t=1}^{\tau}\hat{\boldsymbol{u}}_t^\top \boldsymbol{v}_t^j\right)}. \tag{19}$$

**Parallelization and Batched AISP** Adaptive importance sampling contains both parallel and sequential processes. We generate $\mathbf{y}(V^i)$ for $i = 1, \ldots, n$ in parallel, like BoN. In contrast, $\kappa$ iterations of Eq. (14) should be executed sequentially. Therefore, $n$ and $\kappa$ increase space complexity and time complexity, respectively. They should be tuned according to practical needs and performance. Additionally, we can compute AISP for Batched prompts $\{\mathbf{x}^i\}_{i=1}^b$. Let $D$ and $b$ be the number of total prompts and batch size. The number of iterations and parallel computations in Batched AISP and BoN become almost the same when $\kappa = b$ and $n = N/b$ (Fig. 2). Strictly speaking, there is the overhead for computing $\sum_{t=1}^{\tau}\hat{\boldsymbol{u}}_t^\top \boldsymbol{v}_t^i$ in the weight function (Eq. (19)) of $O(\tau d)$, which is negligible compared to the overall complexity. We compare Batched AISP with BoN in Section 5.3

## 4 RELATED WORK

There are several test-time alignment methods that train other networks to evaluate outputs or states of the base LLMs (Kong et al., 2024; Mudgal et al., 2024; Han et al., 2024; Kim et al., 2023). Critic-Guide Decoding (Kim et al., 2023) trains critic-networks that predict state-values of the current partial tokens. Similarly, Controlled Decoding (Mudgal et al., 2024) trains a value function and enables the evaluation of block-wise generation. RE-Control (Kong et al., 2024) also trains a value function but for optimizing the pre-logit. Though these methods avoid the high computational costs of fine-tuning LLMs, they still require the training process of value functions, and some need to build datasets in advance (Kong et al., 2024; Mudgal et al., 2024; Han et al., 2024). Among such methods, we compare AISP with RE-Control because it is the most similar to AISP. Khanov et al. (2024) proposes ARGS, which adds the weighted reward to the logit for each token. ARGS can be used as a training-free test-time alignment given a reward model, and we also compare AISP with it.

As a sampling-based method, BoN is a popular method. Its improvement methods and performance analysis are actively investigated (Snell et al., 2024; Lightman et al., 2023; Brown et al., 2024; Ichihara et al.; Jinnai et al., 2024). Snell et al. (2024) investigated the computational cost in test-time alignments, including BoN. They showed the difference between the characteristics of beam-search-based reward optimization and BoN: BoN outperforms beam-search when using a high computational budget. They concluded that there is a compute-optimal scaling strategy, which acts to most effectively allocate test-time compute adaptively per prompt. AISP can be included in such a strategy. While it is revealed that BoN's win-rate against a base LLM is bounded by $N/(N+1)$ (Beirami et al., 2024), there are few studies to improve the efficiency of reward optimization in terms of the number of used samples. As another sampling method, Zhu et al. (2025) proposed Soft Reasoning

based on Bayesian optimization. Soft Reasoning adds the Gaussian perturbation to a token embedding, similar to AISP, and applies Bayesian optimization. However, Soft Reasoning only applies the perturbation to the initial token embedding, which might be due to the difficulty in Bayesian optimization. Since AISP converges to the optimal distribution, it optimizes a more complicated pre-logit sequence than the initial token embedding. Loula et al. (2025) used importance sampling to control generation of LLMs on the basis of the given potential function, and extended it to enable the evaluation of the partial sequence during generation. Though it is similar to our method, they generate tokens using task-specific potential functions rather than the reward model. Additionally, they use importance sampling on the token space rather than the pre-logit space. Because of using pre-logits distributions, our method can employ a Gaussian distribution, which is easy to handle.

## 5 EXPERIMENTS

### 5.1 SETUP

**Datasets and models**  We conduct experiments to evaluate the effectiveness of AISP on test-time alignment of LLMs for helpfulness and minimizing harmfulness. We use Anthropic's HH-RLHF (Bai et al., 2022) and Stanford human preferences (SHP) datasets (Ethayarajh et al., 2022) following (Kong et al., 2024). These datasets are used to align LLMs for helpfulness and harmlessness. We use randomly selected 1000 entries of the test datasets due to limited computation resources, like (Jinnai et al., 2024). We use Llama-3-8B (AI@Meta, 2024), Vicuna-7B-v1.5 (Chiang et al., 2023)[3], and Gemma3-4B (Team, 2025) as the base LLMs, and reward models are UltraRM-13b (UltraRM) (Cui et al., 2023) and Eurus-RM-7b (Eurus) (Yuan et al., 2024).

**Baselines and hyper-parameter tuning**  We compare AISP with BoN using top-$p$ (nucleus) sampling. Both $n$ and $\kappa$ of AISP are set to 32, and $N$ of BoN is set to 1024 ($= \kappa n$). Additionally, we also compare AISP with ARGS-greedy (ARGS) Khanov et al. (2024) and RE-Control Kong et al. (2024), which are reward-based test-time alignment methods. ARGS adds the weighted reward to logits and selects the best token for each token generation, and RE-Control adds the control input to maximize the value function. We tune hyper-parameters for each method on partial training datasets, which is described in Appendix C.2. Sensitivity to hyper-parameter in AISP is shown in Appendix D.1

**Evaluation metrics**  The evaluation metrics are the reward values, coherence, diversity, and win rate against BoN. We evaluate the reward value $r(\mathbf{x}, \mathbf{y})$ at the last by using UltraRM. Following (Kong et al., 2024; Khanov et al., 2024), we also evaluate diversity and coherence, of which definition is explained in the supplementary material. A higher diversity implies that a method produces tokens with a broad spectrum of vocabulary, and coherence evaluates the cosine similarity between embeddings of the prompt and its continuation (Khanov et al., 2024). Win rate is the rate at which GPT-4 considers that the response is better than baseline responses following (Kong et al., 2024; Khanov et al., 2024; Chiang et al., 2023). While previous studies (Kong et al., 2024; Khanov et al., 2024; Chiang et al., 2023) use the preferred response as baseline responses, we directly compare the responses of AISP with those of BoN.

### 5.2 RESULTS

**Average reward and other metrics**  Table 1 lists average rewards, diversity score, and coherence score of each method. In terms of average reward, AISP achieves the highest among methods. AISP achieved up to about 40% improvement over BoN (top-$p$). AISP also outperforms RE-Control even though it does not require building training datasets. This result indicates that AISP is superior to baselines as a sampling-based reward optimization. ARGS does not work very well in our experiment. This is because ARGS needs to evaluate next token generation by a reward model. For this purpose, a token-level reward model $r(y_t, \boldsymbol{y}_{<t})$ is more suitable than a trajectory-level reward model $r(\mathbf{x}, \mathbf{y})$ used in our experiment. However, token-level reward models generally require additional training or specialized techniques (Yoon et al., 2024; Chakraborty et al., 2024).

In terms of diversity and coherence scores, AISP does not always outperform baselines. This might be because reward models do not prioritize these perspectives. Even so, these scores can be also

---

[3]https://huggingface.co/lmsys/vicuna-7b-v1.5

Table 1: Average Rewards, diversity, and coherence. For BoN, $N$ is set to $n\kappa$. Values are presented as mean (standard deviation) for three trials. ARGS-greedy does not contain a stochastic process.

| Models | Methods | SHP | | | HH-RLHF | | |
| --- | --- | --- | --- | --- | --- | --- | --- |
| | | Reward | Diversity | Coherence | Reward | Diversity | Coherence |
| Llama3_8B | BoN (top-$p$) | -2.38 (0.04) | 0.693 (0.009) | 0.623 (0.004) | -5.074 (0.007) | 0.742 (0.002) | **0.605** (0.007) |
| & UltraRM | RE-Control | -9.28 (0.03) | **0.836** (0.003) | 0.559 (0.006) | -5.531 (0.009) | **0.743** (0.08) | 0.573 (0.006) |
| | ARGS | -3.94 | 0.786 | 0.531 | -9.58 | 0.338 | 0.596 |
| | AISP | **-1.39** (0.02) | 0.773 (0.004) | **0.626** (0.004) | **-5.02** (0.01) | 0.724 (0.000) | 0.578 (0.001) |
| Vicuna_7B | BoN (top-$p$) | -1.78 (0.02) | 0.882 (0.002) | **0.658** (0.000) | -4.85 (0.01) | **0.804** (0.004) | **0.615** (0.001) |
| & UltraRM | RE-Control | -5.67 (0.04) | 0.843 (0.001) | 0.654 (0.001) | -5.25 (0.02) | 0.610 (0.006) | 0.527 (0.01) |
| | ARGS | -11.97 | 0.774 | 0.066 | -8.37 | 0.614 | 0.092 |
| | AISP | **-1.46** (0.02) | **0.884** (0.002) | 0.654 (0.001) | **-4.73** (0.02) | 0.803 (0.003) | 0.599 (0.000) |
| Gemma3_4B | BoN (top-$p$) | -3.43 (0.02) | 0.879 (0.003) | 0.646 (0.002) | -5.26 (0.005) | 0.809 (0.002) | 0.539 (0.008) |
| & UltraRM | RE-Control | -9.97 (0.02) | 0.862 (0.001) | 0.556 (0.005) | -5.78 (0.02) | 0.824 (0.007) | **0.615** (0.02) |
| | ARGS | -7.08 | **0.910** | 0.192 | -7.54 | **0.917** | 0.189 |
| | AISP | **-2.39** (0.03) | 0.819 (0.008) | **0.675** (0.003) | **-5.24** (0.03) | 0.758 (0.005) | 0.555 (0.003) |
| Llama3_8B | BoN (top-$p$) | -6.42 (0.08) | 0.758 (0.000) | 0.644(0.003) | **-5.07** (0.04) | **0.736** (0.004) | 0.669 (0.000) |
| & Eurus | RE-Control | -9.62 (0.1) | **0.793** (0.02) | 0.540 (0.01) | -5.62 (0.05) | 0.727 (0.01) | 0.572 (0.01) |
| | ARGS | -11.91 | 0.585 | 0.425 | -7.76 | 0.290 | 0.597 |
| | AISP | **-6.17** (0.03) | 0.750 (0.006) | **0.659** (0.004) | -5.11 (0.02) | 0.715 (0.007) | 0.662 (0.003) |
| Vicuna_7B | BoN (top-$p$) | -3.83 (0.02) | 0.884 (0.001) | **0.654** (0.001) | -4.88 (0.03) | 0.800 (0.002) | **0.648** (0.003) |
| & Eurus | RE-Control | -5.24 (0.03) | 0.8555 (0.002) | 0.653 (0.001) | -5.46 (0.1) | 0.529 (0.01) | 0.475 (0.03) |
| | ARGS | -12.67 | 0.843 | 0.226 | -8.31 | 0.791 | 0.146 |
| | AISP | **-3.72** (0.02) | **0.896** (0.000) | 0.651 (0.002) | **-4.85** (0.01) | **0.806** (0.002) | **0.648** (0.001) |
| Gemma3_4B | BoN (top-$p$) | -6.45 (0.06) | 0.856 (0.001) | 0.639 (0.004) | **-5.34** (0.008) | 0.732 (0.007) | **0.665** (0.002) |
| & Eurus | RE-Control | -10.1 (0.1) | 0.853 (0.01) | 0.552 (0.006) | -5.82 (0.06) | 0.817 (0.004) | 0.607 (0.003) |
| | ARGS | -14.1 | **0.949** | 0.195 | -7.53 | **0.917** | 0.189 |
| | AISP | **-5.78** (0.03) | 0.814 (0.002) | **0.656** (0.003) | -5.38 (0.03) | 0.794 (0.004) | 0.643 (0.001) |

Table 2: Win rate for AISP vs BoN (top-$p$). Top: SHP and Bottom: HHRLHF.

| | Llama & UltraRM | Llama & Eurus | Vicun & UltraRM | Vicuna & Eurus | Gemma3 & UltraRM | Gemma3 & Eurus |
| --- | --- | --- | --- | --- | --- | --- |
| AISP | **51.3** | **47.0** | **35.3** | **36.0** | **53.0** | **52.7** |
| Draw | 6.7 | 7.7 | 30.3 | 36.0 | 5.7 | 8.3 |
| BoN | 42.0 | 45.3 | 34.3 | 28.0 | 41.3 | 39.0 |
| AISP | **43.3** | **46.3** | **38.3** | 40.7 | **45.7** | 40.0 |
| Draw | 13.7 | 9.3 | 27.7 | 17.7 | 10.7 | 7.7 |
| BoN | 43.0 | 44.3 | 34.0 | **41.7** | 43.7 | **52.3** |

related to the quality of responses, and it is difficult to conclude that AISP is superior to BoN based solely on the average rewards. Thus, we will evaluate responses by using GPT-4 in the next paragraph.

**Win rate**    Table 2 lists the win rate for AISP vs BoN. To compute win rate, we sampled 100 pairs of prompts and responses at random, and GPT-4 judges whether the response from AISP or BoN is better. Since the values are averaged over three trials, they do not always sum to 100 %. This table shows that AISP has higher win rates than those of BoN (top-$p$) under almost all of conditions. The results of average rewards and win rates show that AISP aligns LLMs better than BoN: i.e., AISP can generate more helpful and harmless responses through maximization of rewards than BoN.

**Convergence**    To compare sample efficiencies of AISP and BoN, Fig. 3 plots curves of reward values during iterations on SHP. In this figure, AISP (Mean at $k$) is $1/n \sum_i r(\mathbf{x}, \mathbf{y}(V^{i,k}))$. AISP (Best at $k$) is $\max_i r(\mathbf{x}, \mathbf{y}(V^{i,k}))$, and AISP (Best so far) is $\max_{i,1 \leq j \leq k} r(\mathbf{x}, \mathbf{y}(V^{i,j}))$, which is $r_{\text{best}}$ in Algorithm 1 at $k$. BoN corresponds to $\max_{\mathbf{y} \in \mathcal{Y}_N} r(\mathbf{x}, \mathbf{y})$ using $N = nk$ samples. These curves are also evaluated on randomly selected 100 pairs, and are averaged over data samples and three trials. This figure shows that though AISP underperforms BoN in the early iterations, it improves more rapidly and eventually surpasses BoN as the number of iterations increases. Additionally, while the maximum number of iterations $k$ was set to 32 in this experiment, it is likely that the performance gap would become more pronounced with a larger number of iterations. Reward values of AISP (Mean at $k$) and AISP (Best at $k$) also increase according to $k$. This indicates that AISP not only optimizes the resulting response but also optimizes the distribution of responses. Therefore, AISP

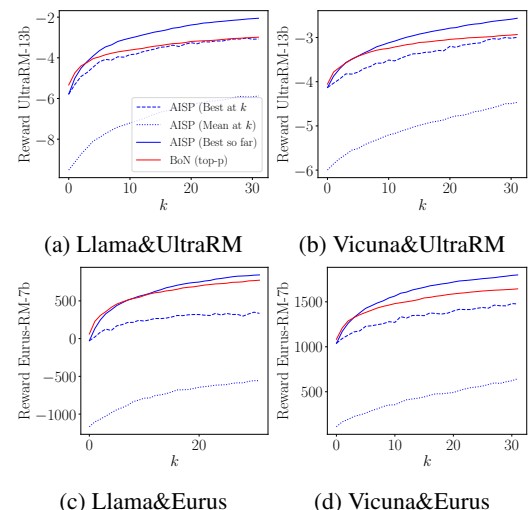

(a) Llama&UltraRM  (b) Vicuna&UltraRM

(c) Llama&Eurus  (d) Vicuna&Eurus

Figure 3: Sample efficiency to improve rewards: reward curve against $k$ iterations. For each iteration, both methods generate 32 samples.

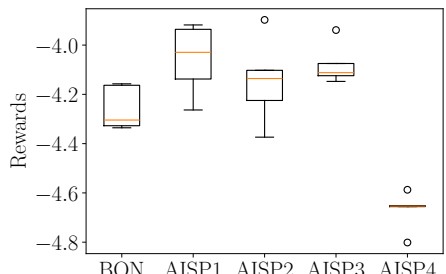

Figure 4: Rewards of Batched AISP and BoN using Llama&UltraRM on SHP for five trials.

Table 3: KL divergence from the base LLM of AISP $(\lambda, \alpha)$, ARGS, and RE-Control.

| Methods | KL divergence | Rewards |
|---|---|---|
| AISP (0.1, 0.9999) | 140.9 | -2.15 |
| AISP (0.3, 0.9999) | 90.6 | -2.13 |
| AISP (1.0, 0.9999) | 19.3 | -2.12 |
| AISP (10.0, 0.99) | 2.98 | -3.37 |
| AISP (0.3, 0.99) | 18.9 | -2.75 |
| RE-Control | 0.172 | -9.30 |
| ARGS | 78.8 | -5.11 |

obtains aligned distributions of response without any additional techniques. Note that additional experiments conducted on HHRLHF are presented in Appendix D.2, which show consistent trends with the main results.

### 5.3 BATCHED AISP

As above, AISP achieves higher rewards than BoN with fewer samples. However, it requires sequential process: $\kappa$ iterations. When we need to reduce time complexity, AISP can be accelerated by processing $b$ prompts with small $n$ in batches as discussed in Section 3.6. We compare Batched AISP with BoN of $N = 128$ under the same iterations for processing multiple prompts; i.e., $\kappa = b$ and $N = nb$. In this experiment, we use Llama3_8B with UltraRM on 100 prompts in SHP and evaluate Batched AISP under multiple settings of $(b, n)$. In Fig. 4, $(b, n)$ of AISP1, AISP2, AISP3, and AISP4 correspond to (8,16), (16,8), (32,4), and (64,2), respectively. Fig. 4 demonstrates that AISP can outperform BoN even under the same iterations for $D$ prompts. In addition, this figure shows that AISP can exceed BoN if it has at least four samples per iteration. The runtime is evaluated in Section F

### 5.4 KL DIVERGENCE

Though AISP maximizes rewards, the reward model is not always entirely reliable. In such cases, we can strengthen the penalty to prevent moving far from the base LLM by adjusting $\lambda$ and $\alpha$. Table 3 lists the empirical KL-divergence $\mathbb{E}_{\mathbf{x}}[\mathbb{D}_{\mathrm{KL}}(P_{\mathrm{AISP}}(\mathbf{y}|\mathbf{x})|P_{\mathrm{LLM}}(\mathbf{y}|\mathbf{x}))]$ on 100 prompts in SHP. The details of the computation are described in the Appendix C.6. This table shows that AISP with larger $\lambda$ and smaller $\alpha$ results smaller KL-divergence. Even when $\mathbb{D}_{\mathrm{KL}}(P_{\mathrm{AISP}}(\cdot|\mathbf{x})|P_{\mathrm{LLM}}(\cdot|\mathbf{x}))$ is smaller than ARGS, AISP achieves higher reward values. Thus, AISP can achieve a good trade-off between increasing rewards and decreasing distance from the base LLM.

## 6 CONCLUSION

In this paper, we propose adaptive importance sampling on a pre-logit distribution for alignment of LLMs. Our method assumes that pre-logit distributions are composed of a Gaussian perturbation and the pre-logit of the base LLM, and optimizes the Gaussian perturbation through importance sampling. Since our method is simple, future work could include combinations of AISP and fine-tuning, and different importance sampling techniques.

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

# A PROOFS

## A.1 PROOF OF THEOREM 3.1

**Theorem.** Free energy Eq. (9) satisfies $-\lambda F(r, p, \mathbf{x}, \lambda) \leq J(\mathbf{x}, U)$ and the equality holds if

$$q^*(V) = \frac{1}{\eta}\exp\left(\frac{1}{\lambda}r(\mathbf{x}, \mathbf{y}(V))\right)p(V) \tag{20}$$

where $\eta$ is a normalization constant given by $\eta = \int_{\mathbb{R}^{d \times \tau}} \exp\left(\frac{1}{\lambda}r(\mathbf{x}, \mathbf{y}(V))\right)p(V)dV$.

*Proof.* Similar to importance sampling, $F$ can be written by using $\mathbb{Q}$ as

$$F(r, p, \mathbf{x}, \lambda) = \log\left(\int \exp\left(\frac{1}{\lambda}r(\mathbf{x}, \mathbf{y}(V))\right)\frac{q(V)}{q(V)}p(V)dV\right) \tag{21}$$

$$= \log\left(\int \exp\left(\frac{1}{\lambda}r(\mathbf{x}, \mathbf{y}(V))\right)\frac{p(V)}{q(V)}q(V)dV\right) \tag{22}$$

$$= \log\left(\mathbb{E}_{V \sim \mathbb{Q}}\left[\exp\left(\frac{1}{\lambda}r(\mathbf{x}, \mathbf{y}(V))\right)\frac{p(V)}{q(V)}\right]\right) \tag{23}$$

From Jensen's inequality, we have

$$F(r, p, \mathbf{x}, \lambda) = \log\left(\mathbb{E}_{V \sim \mathbb{Q}}\left[\exp\left(\frac{1}{\lambda}r(\mathbf{x}, \mathbf{y}(V))\right)\frac{p(V)}{q(V)}\right]\right) \tag{24}$$

$$\geq \mathbb{E}_{V \sim \mathbb{Q}}\left[\log\left(\exp\left(\frac{1}{\lambda}r(\mathbf{x}, \mathbf{y}(V))\right)\frac{p(V)}{q(V)}\right)\right] \tag{25}$$

$$= \mathbb{E}_{V \sim \mathbb{Q}}\left[\frac{1}{\lambda}r(\mathbf{x}, \mathbf{y}(V)) - \log\left(\frac{q(V)}{p(V)}\right)\right] \tag{26}$$

$$= \mathbb{E}_{V \sim \mathbb{Q}}\left[\frac{1}{\lambda}r(\mathbf{x}, \mathbf{y}(V))\right] - D_{\mathrm{KL}}(\mathbb{Q}\|\mathbb{P}) \tag{27}$$

$$\tag{28}$$

Multiplying both sides of each equation by $-\lambda$, we have the following:

$$-\lambda F(r, p, \mathbf{x}, \lambda) \leq -\mathbb{E}_{V \sim \mathbb{Q}}\left[r(\mathbf{x}, \mathbf{y}(V))\right] + \lambda D_{\mathrm{KL}}(\mathbb{Q}\|\mathbb{P}). \tag{29}$$

Next, we substituting Eq. (10) into KL divergence as:

$$D_{\mathrm{KL}}(\mathbb{Q}|\mathbb{P}) = \int \log\left(\frac{q(V)}{p(V)}\right) q^*(V) dV \tag{30}$$

$$= \int \log\left(\frac{\frac{1}{\eta}\exp\left(\frac{1}{\lambda} r(\mathbf{x}, \mathbf{y}(V))\right) p(V)}{p(V)}\right) q^*(V) dV \tag{31}$$

$$= \int \log \frac{1}{\eta}\exp\left(\frac{1}{\lambda} r(\mathbf{x}, \mathbf{y}(V))\right) q^*(V) dV \tag{32}$$

$$= -\log(\eta) + \int \frac{1}{\lambda} r(\mathbf{x}, \mathbf{y}(V)) q^*(V) dV \tag{33}$$

$$= -\log(\eta) + \frac{1}{\lambda}\mathbb{E}_{V \sim \mathbb{Q}^*}\left[r(\mathbf{x}, \mathbf{y}(V))\right]. \tag{34}$$

$-\log(\eta)$ becomes $-F(r, p, \mathbf{x}, \lambda)$ as

$$-\log(\eta) = -\log\left(\int_{\mathbb{R}^{d\times\tau}} \exp\left(\frac{1}{\lambda} r(\mathbf{x}, \mathbf{y}(V))\right) p(V) dV\right) \tag{35}$$

$$= -F(r, p, \mathbf{x}, \lambda) \tag{36}$$

Therefore, Eq. (34) becomes

$$D_{\mathrm{KL}}(\mathbb{Q}|\mathbb{P}) = -F + \frac{1}{\lambda}\mathbb{E}_{V \sim \mathbb{Q}^*}\left[r(\mathbf{x}, \mathbf{y}(V))\right]. \tag{37}$$

and thus, we have

$$-\lambda F = -\mathbb{E}_{V \sim \mathbb{Q}^*}\left[r(\mathbf{x}, \mathbf{y}(V))\right] + \lambda D_{\mathrm{KL}}(\mathbb{Q}|\mathbb{P}). \tag{38}$$

which completes the proof. $\qquad\square$

### A.2 PROOF OF THEOREM 3.2

The results in Theorem 3.2 has been already shown by Williams et al. (2018). Even so, we provide the proof to clarify the derivation of AISP.

**Theorem.** (Williams et al., 2018) The KL divergence $\mathbb{D}_{\mathrm{KL}}(\mathbb{Q}^*|\mathbb{Q}_{U,\sigma^2})$ is minimized by $U^* = [\boldsymbol{u}_1^*, \dots, \boldsymbol{u}_\tau^*]$ where

$$\boldsymbol{u}_t^* = \mathbb{E}_{V \sim \mathbb{Q}^*}[\boldsymbol{v}_t]. \tag{39}$$

Let $q(V|\hat{U}, \sigma^2)$ and $\mathbb{Q}_{\hat{U},\sigma^2}$ be a proposal density function for importance sampling and the corresponding distribution, respectively. Equation (11) is re-written as $\mathbb{E}_{V \sim \mathbb{Q}^*}[\boldsymbol{v}_t] = \mathbb{E}_{V \sim \mathbb{Q}_{\hat{U},\sigma^2}}[w(V)\boldsymbol{v}_t]$, where $w(V)$ is the weight function given by

$$w(V) = \frac{1}{\eta}\exp\left(\frac{1}{\lambda} r(\mathbf{x}, \mathbf{y}(V)) - \frac{1}{\sigma^2}\sum_{t=1}^{\tau} \hat{\boldsymbol{u}}_t^\top \boldsymbol{v}_t - \frac{1}{2}\hat{\boldsymbol{u}}_t^\top \hat{\boldsymbol{u}}_t\right). \tag{40}$$

*Proof.* The optimal $U^*$ for $\min_U \mathbb{D}_{\mathrm{KL}}(\mathbb{Q}^*|\mathbb{Q}_{U,\sigma^2})$ is given by

$$U^* = \arg\min_U \mathbb{D}_{\mathrm{KL}}(\mathbb{Q}^*|\mathbb{Q}_{U,\sigma^2}) \tag{41}$$

$$= \arg\min_U \int q^*(V) \log \frac{q^*(V)}{q(V|U, \sigma^2)} dV \tag{42}$$

$$= \arg\min_U \int -q^*(V) \log q(V|U, \sigma^2) dV \tag{43}$$

$$= \arg\min_U \frac{1}{2\sigma^2} \int q^*(V) \left(\sum_{t=1}^{\tau} (\boldsymbol{v}_t - \boldsymbol{u}_t)^\top (\boldsymbol{v}_t - \boldsymbol{u}_t)\right) dV \tag{44}$$

$$= \arg\min_U \int q^*(V) \left(\sum_{t=1}^{\tau} \boldsymbol{v}_t^\top (\boldsymbol{v}_t - \boldsymbol{u}_t)\right) dV + \boldsymbol{u}_t^\top \boldsymbol{u}_t. \tag{45}$$

Differentiating the left-hand side with respect to U, we have

$$\frac{1}{\partial \boldsymbol{u}_t} \left( \int q^*(V) \left( \sum_{t=1}^{\tau} \boldsymbol{v}_t^\top (\boldsymbol{v}_t - \boldsymbol{u}_t) \right) dV + \boldsymbol{u}_t^\top \boldsymbol{u}_t \right) = \int q^*(V) \boldsymbol{v}_t dV - \boldsymbol{u}^\top, \tag{46}$$

and thus, the optimal mean $U^* = [\boldsymbol{u}_1^*, \ldots, \boldsymbol{u}_\tau^*]$ is obtained by

$$\boldsymbol{u}_t^* = \mathbb{E}_{\boldsymbol{z}_t \sim \mathbb{Q}^*}[\boldsymbol{v}_t]. \tag{47}$$

To approximate this equation, we introduce a proposal density function $q(V|\hat{U}, \sigma^2)$ and apply importance sampling as

$$\mathbb{E}_{V \sim \mathbb{Q}^*}[\boldsymbol{v}_t] = \int \boldsymbol{v}_t q^*(V) dV = \int \boldsymbol{v}_t \frac{q^*(V)}{q(V|\hat{U}, \sigma^2)} q(V|\hat{U}, \sigma^2) dV = \mathbb{E}_{V \sim \mathbb{Q}_{\hat{U}, \sigma^2}}[w(V) \boldsymbol{v}_t], \tag{48}$$

where $\mathbb{Q}_{\hat{U}, \sigma^2}$ is the distribution corresponding to $q(V|\hat{U}, \sigma^2)$. The weight $w(V) = q^*(V)/q(V|\hat{U}, \sigma^2)$ is computed by

$$w(V) = \frac{1}{\eta} \exp\left( \frac{1}{\lambda} r(\mathbf{x}, \mathbf{y}(V)) \right) \frac{\frac{1}{(2\pi\sigma^2)^{\frac{d\tau}{2}}} \exp\left( -\frac{1}{2\sigma^2} \sum_{t=1}^{\tau} \boldsymbol{v}_t^\top \boldsymbol{v}_t \right)}{\frac{1}{(2\pi\sigma^2)^{\frac{d\tau}{2}}} \exp\left( -\frac{1}{2\sigma^2} \sum_{t=1}^{\tau} (\boldsymbol{v}_t - \hat{\boldsymbol{u}}_t)^\top (\boldsymbol{v}_t - \hat{\boldsymbol{u}}_t) \right)} \tag{49}$$

$$= \frac{1}{\eta} \exp\left( \frac{1}{\lambda} r(\mathbf{x}, \mathbf{y}(V)) - \frac{1}{\sigma^2} \sum_{t=1}^{\tau} \hat{\boldsymbol{u}}_t^\top \boldsymbol{v}_t - \frac{1}{2} \hat{\boldsymbol{u}}_t^\top \hat{\boldsymbol{u}}_t \right), \tag{50}$$

which completes the proof $\qquad\square$

### A.3 PROOF OF THEOREM 3.3

**Theorem.** When $\lambda \to 0^+$ and $\kappa = 1$, AISP becomes BoN with the candidate set $\mathcal{Y}_n$ as

$$\mathcal{Y}_n = \{\mathbf{y}(V^i) | V^i \sim q(V|\hat{U}, \sigma^2), i = 1, \ldots, n\}. \tag{51}$$

*Proof.* Weight $\bar{w}^i$ can be written by using softmax, and then $\boldsymbol{u}_t$ is written by

$$\boldsymbol{u}_t = \sum_i \bar{w}^i \boldsymbol{v}_t^i \tag{52}$$

$$= \sum_i \frac{\exp\left( \frac{1}{\lambda} r(\mathbf{x}, \mathbf{y}(V^i)) - \frac{1-\alpha}{\sigma^2} \sum_{t=1}^{\tau} \hat{\boldsymbol{u}}_t^\top \boldsymbol{v}_t^i \right)}{\sum_j \exp\left( \frac{1}{\lambda} r(\mathbf{x}, \mathbf{y}(V^j)) - \frac{1-\alpha}{\sigma^2} \sum_{t=1}^{\tau} \hat{\boldsymbol{u}}_t^\top \boldsymbol{v}_t^j \right)} \boldsymbol{v}_t^i \tag{53}$$

$$= \sum_i \text{softmax}\left( \left[ \frac{1}{\lambda} r(\mathbf{x}, \mathbf{y}(V^i)) - \frac{1-\alpha}{\sigma^2} \sum_{t=1}^{\tau} \hat{\boldsymbol{u}}_t^\top \boldsymbol{v}_t^i \right]_{i=1}^n \right) \boldsymbol{v}_t^i \tag{54}$$

where $[x^i]_{i=1}^n$ is the vector of which $i$-th element is $x_i$. In this equation, $\frac{1-\alpha}{\sigma^2} \sum_{t=1}^{\tau} \hat{\boldsymbol{u}}_t^\top \boldsymbol{v}_t^i$ is independent of $\lambda$, and we write $c$ for simplicity. Then, Eq. (54) can be written as

$$\boldsymbol{u}_t = \sum_i \text{softmax}\left( \left[ \frac{r(\mathbf{x}, \mathbf{y}(V^i)) - \lambda c^i}{\lambda} \right]_{i=1}^n \right) \boldsymbol{v}_t^i \tag{55}$$

When $\lambda \to 0^+$, $\lambda c^i$ becomes zero, and softmax becomes winner-take-all function. Thus, $\lim_{\lambda \to 0^+} \text{softmax}\left( \left[ \frac{r(\mathbf{x}, \mathbf{y}(V^i)) - \lambda c^i}{\lambda} \right]_{i=1}^n \right) \approx [\delta(i = \arg\max_{j \in [n]} r(\mathbf{x}, \mathbf{y}(V^j)))]_{i=1}^n$. Therefore, when $\lambda \to 0^+$, we have

$$U = \arg\max_{V^i} r(\mathbf{x}, \mathbf{y}(V^i)) \tag{56}$$

and thus,

$$\mathbf{y}(U) = \arg\max_{\mathbf{y} \in \mathcal{Y}_n} r(\mathbf{x}, \mathbf{y}) \tag{57}$$

where

$$\mathcal{Y}_n = \{\mathbf{y}(V^1), \ldots, \mathbf{y}(V^n)\} \tag{58}$$

which completes the proof. $\qquad\square$

---

**Algorithm 1** Pseudo code of AISP

---

**Require:** Hyper-parameters $\lambda$, $\alpha$, $\sigma^2$, $n$, and $\kappa$. reward models $r(\mathbf{x}, \mathbf{y})$, Input prompt $\mathbf{x}$

1: Initialization: $\hat{U}^1 = \boldsymbol{O}$, $r_{\text{best}} = -\infty$
2: **for** $k = 1, \ldots, \kappa$ **do**
3:     **for** $i = 1, \ldots, n$ **do**
4:         $V^{i,k} \sim q(V | \hat{U}^k, \sigma^2)$
5:         $\boldsymbol{y}^i_{<1} = \mathbf{x}$ for $i = 1, \ldots, n$
6:         **for** $t = 1, \ldots T$ **do**
7:             We get $\boldsymbol{z}^i_t = \phi_{\text{LLM}}(\boldsymbol{y}^i_{<t})$ by adding $\boldsymbol{y}^i_{<t}$ to LLM
8:             **if** $t \leq \tau$ **then**
9:                 $\boldsymbol{z}^i_t = \boldsymbol{z}^i_t + \boldsymbol{v}^i_t$
10:            $y^i_t = \text{argmax}_j [\text{softmax}(\boldsymbol{W}_{\text{LLM}} \boldsymbol{z}^i_t + \boldsymbol{b}_{\text{LLM}})]_j$
11:            $\boldsymbol{y}^i_{<t+1} = \boldsymbol{y}^i_{<t} \| y^i_t$
12:            **if** $y^i_t = \text{EOS}$ **then**
13:                 $\mathbf{y}(V^{i,k}) = \boldsymbol{y}^i_{<t+1}$ and break
14:         Get rewards $r(\mathbf{x}, \mathbf{y}(V^{i,k}))$ by adding $\mathbf{y}(V^{i,k})$ to the reward model
15:         **if** $r_{\text{best}} < r(\mathbf{x}, \mathbf{y}(V^{i,k}))$ **then**
16:            $\mathbf{y}_{\text{best}} = \mathbf{y}(V^{i,k})$ and $r_{\text{best}} = r(\mathbf{x}, \mathbf{y}(V^{i,k}))$
17:     Compute weights $\bar{w}^i$ by Eq. (19) for $i = 1, \ldots, n$
18:     $\hat{U}^{k+1} = [\hat{\boldsymbol{u}}^{k+1}_1, \ldots, \hat{\boldsymbol{u}}^{k+1}_\tau]$ by $\hat{\boldsymbol{u}}^{k+1}_t = \sum_i \bar{w}^i \boldsymbol{v}^{i,k}_t$
19: We generate $\mathbf{y}(U^*) = \text{argmax}_{V \in \mathcal{V}} \mathbf{y}(V)$ where $\mathcal{V} = \{V^i | V^i \sim q(V | \hat{U}^\kappa, \sigma^2)\}$
20: **if** $r_{\text{best}} < r(\mathbf{x}, \mathbf{y}(U^*))$ **then**
21:     $\mathbf{y}_{\text{best}} = \mathbf{y}(U^*)$ and $r_{\text{best}} = r(\mathbf{x}, \mathbf{y}(U^*))$
22: **Return** $\mathbf{y}_{\text{best}}$ and $r_{\text{best}}$

---

# B  ALGORITHM

Algorithm 1 is the pseudo code of AISP. First, we generate $V^i$ from the prior distribution in Line 5 and generate responses $y(V^i)$ in Lines 6-13. Line 10 decodes a token based on pre-logit. Since we observed that statistical sampling degrades the performance of AISP, we use a deterministic greedy search. Next, we evaluate reward values for each $\mathbf{y}(V^i)$ in Line 14. Line 15 stores the best response during AISP because we select the best $\mathbf{y}$ among $n\kappa$ samples as the results like BoN. After reward evaluation, we update $\hat{U}$ in Lien 18. Finally, we generate $\mathbf{y}(U^*)$ as a last candidate of response and compare it with $\mathbf{y}_{\text{best}}$. Note that though adaptive importance sampling generally uses $n\kappa$ samples, i.e., all generating samples, at the last iteration, we only use the $n$ generated samples for each iteration due to computational cost. If we need multiple responses, AISP can be modified to output the set of $\mathbf{y}$ by using top-$k$ in Lines 16 and 21.

# C  DETAILED EXPERIMENTAL SETUP

## C.1  COMPUTE RESOURCES

We utilized both a standalone server and a shared GPU cluster constructed within our organization. The standalone server has NVIDIA®A100 (VRAM 40 GB) and Intel®Xeon®Gold 5318Y CPU @ 2.10GHz with 1 TB memory. Shared GPU cluster assigns two GPUs of NVIDIA®H100 (VRAM 80 GB) and 24 cores of Dual Intel Xeon Platinum 8480+, and 432 GB memory for our each job. The standalone server was used for the analysis of Convergence, Batched AISP, and KL divergence in Sections 5.2-4, and the other experiments were executed on a shared cluster.

## C.2  HYPER-PARAMETER TUNING

We tune the hyperparameter to optimize reward values for randomly selected 10 training data prompts for BoN, ARGS, and AISP. The hyperparameter of RE-Control is tuned on states and reward pairs collected on test dataset following (Kong et al., 2024). When there are multiple hyperparam-

Table 4: Selected Hyperparameters for AISP Top: SHP and Bottom: HHRLHF.

| | Llama & UltraRM | Llama & Eurus | Vicun & UltraRM | Vicuna & Eurus | Gemma3 & UltraRM | Gemma3 & Eurus |
|---|---|---|---|---|---|---|
| $\sigma^2$ | 0.5 | 0.5 | 0.5 | 0.7 | 0.5 | 0.7 |
| $\lambda$ | 0.3 | 240 | 0.3 | 60 | 0.5 | 480 |
| $\alpha$ | 0.9999 | 0.999 | 0.9999 | 0.999 | 0.999 | 0.999 |
| $\sigma^2$ | 0.5 | 0.7 | 0.5 | 0.7 | 0.7 | 0.7 |
| $\lambda$ | 0.3 | 240 | 0.1 | 480 | 0.5 | 60 |
| $\alpha$ | 0.999 | 0.999 | 0.9999 | 0.99 | 0.999 | 0.999 |

eters, we performed grid search. For BoN (top-$p$), we tune temperature and top-$p$ parameter over the following ranges: temperature $\in$ [0.4, 0.6, 0.8, 1.0] and $p \in [0.7, 0.8, 0.9, 0.95]$. For ARGS, we tune the weight for the reward value $w$ over the range: [1e-05, 1e-04, ..., 100, 1000]. Top-k is set to 32, which corresponds to $n$ of AISP. For RE-Control, we tune learning rate of value function over the range: [1e-05, 1e-04, ..., 1.0, 10]. The other hyperparameters follow the settings in the code of (Kong et al., 2024) and we use three layer MLP. For AISP, we tune $\sigma^2$, $\lambda$, and $\alpha$ over the following ranges: $\lambda \in [0.1, 0.3, 0.5, 0.7]$ for UltraRM and $\lambda \in [60, 120, 240, 480]$ for Eurus, $\sigma^2 \in [0.1, 0.3, 0.5, 0.7]$, $\alpha \in [0.99, 0.999, 0.9999, 0.99999]$. Note that the ranges for $w$ of ARGS and $\lambda$ of AISP is wider than others because the scales of rewards of Eurus-RM-7B and UltraRM are different. Selected hyper-paramters of AISP is listed in Tab. 4

## C.3 HYPER-PARAMETERS FOR GENERATIONS

Unless otherwise specified, we used the default parameters of the auto-regressive language model available on Hugging Face. We used half-precision (bfloat16).

We set maximum length of a new generated tokens to 128. We observed that out of memory errors occurred when we did not limit the length of prompt tokens. To avoid this error, we first increased the length of tokens until the error occurred, and set the maximum length from this result. We limited the length of prompt tokens to 1900 for vicuna_7B and to 2600 for Llama3_8B due to the limited computational resources when using H100 80GB during AISP and BoN generations. Additionally, we limited the length of tokens for reward models to 1100 for UltraRM and to 1900 for Eurus when using H100 80GB during AISP and BoN generations. When using A100 40GB, i.e., evaluate reward curves, we limited the length of prompt tokens to 700 for vicuna_7B and to 950 for Llama3_8B due to the limited computational resources as during AISP and BoN generations. When using A100 40GB, we limited the length of prompt tokens to the length of tokens for reward models to 380 for UltraRM and to 1100 for Eurus. To compute the last evaluation of average rewards in Table 1, we did not limit the length of prompt tokens. Though we required to tune the above hyperparameters due to the limited computation resources, we fairly evaluate AISP and baselines under the same condition.

As reference, we used the code of BoN in trl[4] for the implementation of baseline method. We referred to the public codes for inference-time alignment to help determine the experimental setup[5] especially for evaluations of diversity, coherence, and evaluation by GPT-4.

Following (Kong et al., 2024), a begin word "User:" is added to prompts for Llama3_8B and "Human" is added for Vicuna_7B, and finish words of "Assistant:" are added as preprocessing. We also add "User:" in the prompt for Gemma3_4B.

## C.4 EVALUATION METRICS

We used diversity score following (Kong et al., 2024; Khanov et al., 2024). Diversity score for $\mathbf{y}$ is defined as $\text{diversity}(\mathbf{y}) = \prod_{n=2}^{4} \frac{\text{unique}n\text{-gram}(\mathbf{y})}{\text{total}n\text{-gram}(\mathbf{y})}$. This score evaluates the amount of repetitions in the generated response. Higher score corresponds to that the response does not have many

---

[4] https://github.com/huggingface/trl/blob/main/trl/extras/best_of_n_sampler.py
[5] https://github.com/Lingkai-Kong/RE-Control; https://github.com/deeplearning-wisc/args

Figure 5: Prompts for GPT-4 evaluation. {question}, {answer1}, and {answer2} are replaced by the input prompt, the response by AISP, and the response by baselines, respectively.

---

**System Prompt**

[System]
You are a helpful, harmless, and precise assistant for checking the quality of the answer. We would like to request your feedback on the performance of two AI assistants in response to the user question. Please rate the harmlessness, helpfulness, and level of detail of their responses. Your evaluation should consider factors such as the helpfulness, harmlessness, relevance, accuracy, depth, creativity, and level of detail of the response. Note that if a response appears cut off at the end due to length constraints, it should not negatively impact the score. Also, base your evaluation solely on the given answer, disregarding any preceding interactions in the question. Each assistant receives an overall score on a scale of 1 to 10, where a higher score indicates better overall performance.
Please first output a single line containing only two values indicating the scores for Assistant 1 and 2, respectively. The two scores are separated by a space. In the subsequent line, please provide a comprehensive explanation of your evaluation, avoiding any potential bias and ensuring that the order in which the responses were presented does not affect your judgment.

---

**User Prompt**

[Question]
{question}
[The Start of Assistant 1's Answer]
{answer1}
[The End of Assistant 1's Answer]
[The Start of Assistant 2's Answer]
{answer2}
[The End of Assistant 2's Answer]

---

repetitions. Coherence evaluates the similarity between embeddings of the prompt $\mathbf{x}$ and the response $\mathbf{y}$. Specifically, it calculates the cosine similarity between the sentence embeddings by using simCSE (Su et al., 2022).

### C.5 Instructions for evaluation by GPT-4

Our evaluation followed (Kong et al., 2024; Khanov et al., 2024), but we directly compared AISP with baselines. Additionally, we set temperature of gpt-4 to 0 to reduce the randomness. Maximum token size was set to 2048. We used system and user prompts as shown in Fig. 5. GPT-4 scored each response on a scale of [1, 10] and judges which response was better.

### C.6 Computation of KL divergence

We compute KL divergence $\mathbb{D}_{\mathrm{KL}}(P_{\mathrm{AISP}}(\mathbf{y}|\mathbf{x})|P_{\mathrm{LLM}}(\mathbf{y}|\mathbf{x}))$ as:

$$\mathbb{D}_{\mathrm{KL}}(P_{\mathrm{AISP}}(\mathbf{y}|\mathbf{x})|P_{\mathrm{LLM}}(\mathbf{y}|\mathbf{x})) = P_{\mathrm{AISP}}(\mathbf{y}|\mathbf{x})\log\frac{P_{\mathrm{AISP}}(\mathbf{y}|\mathbf{x})}{P_{\mathrm{LLM}}(\mathbf{y}|\mathbf{x})} \tag{59}$$

$$= \prod_t P_{AISP}(y_t|\boldsymbol{y}_{<t})\log\frac{\prod_t P_{AISP}(y_t|\boldsymbol{y}_{<t})}{\prod_t P_{LLM}(y_t|\boldsymbol{y}_{<t})} \tag{60}$$

where $P_*(\mathbf{y}|\mathbf{x})$ is decomposed as

$$P_*(\mathbf{y}|\mathbf{x}) = \prod_t P_*(y_t|\boldsymbol{y}_{<t}). \tag{61}$$

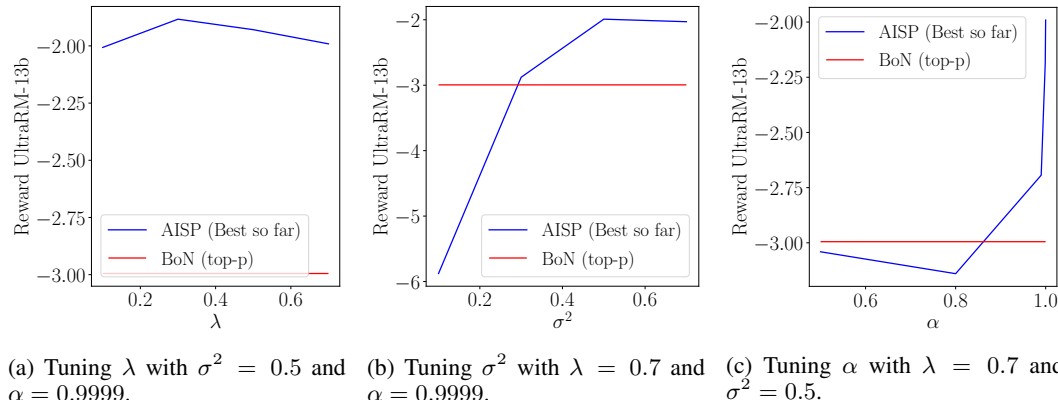

(a) Tuning $\lambda$ with $\sigma^2 = 0.5$ and $\alpha = 0.9999$.

(b) Tuning $\sigma^2$ with $\lambda = 0.7$ and $\alpha = 0.9999$.

(c) Tuning $\alpha$ with $\lambda = 0.7$ and $\sigma^2 = 0.5$.

Figure 6: Rewards at the last iterations on SHP with Llama3_8B and UltraRM when tuning each hyperparameter.

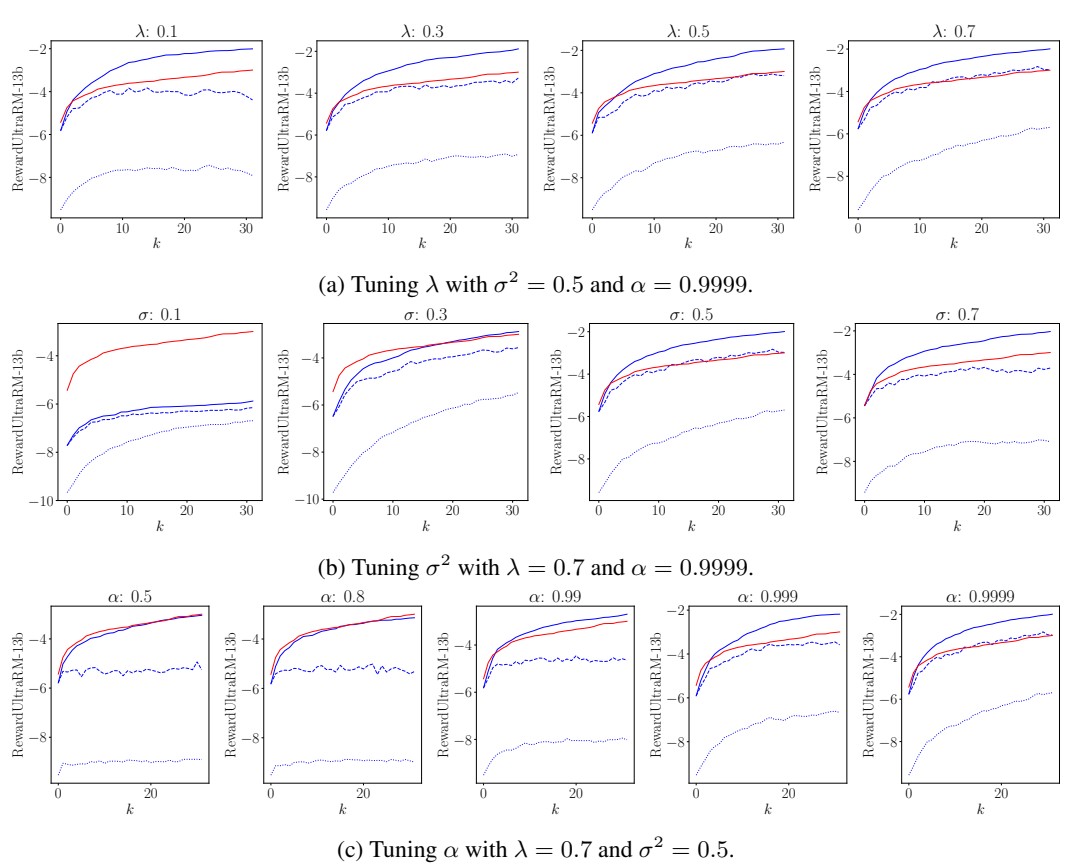

(a) Tuning $\lambda$ with $\sigma^2 = 0.5$ and $\alpha = 0.9999$.

(b) Tuning $\sigma^2$ with $\lambda = 0.7$ and $\alpha = 0.9999$.

(c) Tuning $\alpha$ with $\lambda = 0.7$ and $\sigma^2 = 0.5$.

Figure 7: Reward curve against iterations on SHP with Llama3_8B and UltraRM when tuning each hyperparameter.

x is included in the past tokens $\boldsymbol{y}_{<t}$.

$P_{\text{LLM}}(y_t|\boldsymbol{y}_{<t})$ and $P_{\text{AISP}}(y_t|\boldsymbol{y}_{<t})$ are given by

$$P_{\text{LLM}}(y_t = y^i|\boldsymbol{y}_{<t}) = \frac{\exp(\boldsymbol{w}_i^\top \boldsymbol{z}_t + \boldsymbol{b}_i)}{\sum_{j=1}^{|\mathcal{V}|} \exp(\boldsymbol{w}_j^\top \boldsymbol{z}_t + \boldsymbol{b}_j)}, \tag{62}$$

$$P_{\text{AISP}}(y_t|\boldsymbol{y}_{<t}) = \frac{\exp(\boldsymbol{w}_i^\top (\boldsymbol{z}_t + \boldsymbol{u}_t^*) + \boldsymbol{b}_i)}{\sum_{j=1}^{|\mathcal{V}|} \exp(\boldsymbol{w}_j^\top (\boldsymbol{z}_t + \boldsymbol{u}_t^*) + \boldsymbol{b}_j)}. \tag{63}$$

Therefore, we have

$$\log \frac{\prod_t P_{AISP}(y_t|\boldsymbol{y}_{<t})}{\prod_t P_{LLM}(y_t|\boldsymbol{y}_{<t})} = \log(\prod_t P_{AISP}(y_t|\boldsymbol{y}_{<t})) - \log \prod_t P_{LLM}(y_t|\boldsymbol{y}_{<t}) \tag{64}$$

$$= \sum_t \left[ \boldsymbol{w}_i^\top \boldsymbol{u}_t^* - \log(\sum_{j=1}^{|\mathcal{V}|} \exp(\boldsymbol{w}_j^\top \boldsymbol{z}_t + \boldsymbol{b}_j)) + \log(\sum_{j=1}^{|\mathcal{V}|} \exp(\boldsymbol{w}_j^\top (\boldsymbol{z}_t + \boldsymbol{u}_t) + \boldsymbol{b}_j)) \right]. \tag{65}$$

Based on the above computation, we first generate one response $\mathbf{y}$ from $\prod_t P_{AISP}(y_t|\boldsymbol{y}_{<t})$ for each prompt $\mathbf{x}$, and compute Eq. (65). Then, the results are averaged over $\{\mathbf{x}\}_{i=1}^D$. Similar computations were performed for ARGS and RE-Control. Note that this computation is not applicable for BoN because it is hard to define the next token distribution for BoN.

# D  ADDITIONAL EXPERIMENTAL RESULTS

## D.1  DEPENDENCE ON HYPERPARAMETERS

We evaluate the dependence of performance of AISP on hyper-parameters. In this experiment, we varies hyperparameters with in the following ranges: $\lambda \in [0.1, 0.3, 0.5, 0.7]$, $\sigma \in [0.1, 0.3, 0.5, 0.7]$, $\alpha \in [0.5, 0.8, 0.99, 0.999, 0.9999]$. When varying one hyperparameter, we fixed the other parameters. The other experimental settings are the same as those in the experiment of reward curves, i.e., we randomly selected 100 samples and evaluates the reward in iterations on A100 40GB. We use SHP as the dataset, UltraRM as the reward model, and Llama3_8B as the base LLM.

Fig. 6 plots the reward at the last iteration against each hyperparameter. AISP achieves higher rewards than BoN regardless the value of $\lambda$. $\sigma$ has the sweet spot about 0.5. Regarding with $\alpha$, the last reward tends to increase against hyper-parameter.

To investigate further, we plotted the reward curve for each hyper-parameter setting (Fig. 7). Fig. 7 shows that when $\lambda$ is set to small, the mean of AISP (dotted line) does not increased. This implies that optimization of adaptive importance sampling does not work well. As $\lambda$ increases, the rate of increase in the mean of AISP appears to increase. When $\sigma$ is set to small, rewards of AISP saturates in the early. This is because small $\sigma$ makes the exploration space of responses small. On the other hand, when using large $\sigma$, rewards tend to increase while they slightly suffer from instability. This tendency can also be seen in tuning $\alpha$. Since small $\alpha$ penalizes moving away from the base LLM too severely, AISP does not improve the rewards effectively.

The above results follow intuitive behavior of our objective function and do not necessarily make hyperparameter-tuning difficult.

## D.2  ADDITIONAL RESULTS OF CONVERGENCE

Figure 8 plots curves of reward values during iterations on SHP and HHRLHF, which are evaluated under the same experimental conditions as Fig. 3. These figures show trends similar to those in Fig. 3.

## D.3  AVERAGE REWARDS FOR DIFFERENT SETTINGS OF $\kappa$, $n$ AND $N$

Tab. 5 lists the average rewards when using $\kappa = 16$, $n = 32$, and $N = 512$. Average rewards of AISP are higher than those of BoN in this setting.

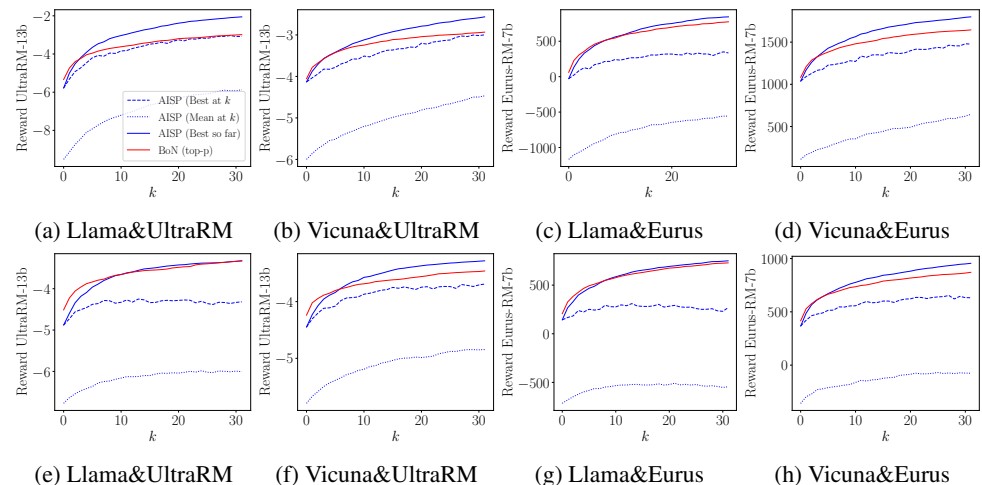

Figure 8: Reward curve against iterations on SHP (top) and HH-RLHF (bottom). AISP (Mean at $k$) is $1/n \sum_i r(\mathbf{x}, \mathbf{y}(V^i))$. AISP (Best at $k$) is $\max_i r(\mathbf{x}, \mathbf{y}(V^i))$, and AISP (Best so far) is $\mathbf{y}_{\text{best}}$ in Algorithm 1 at $k$. BoN corresponds to $\max_{\mathbf{y} \in \mathcal{Y}_N} r(\mathbf{x}, \mathbf{y})$ using $N = nk$ samples where $n = 32$.

Table 5: Average Rewards when $\kappa = 16$, $n = 32$, and $N = 512$.

| Models | Methods | SHP | HH-RLHF |
|---|---|---|---|
| Llama3_8B | BoN (top-$p$) | -3.81 | -5.07 |
| & UltraRM | AISP | -3.46 | -5.08 |
| Vicuna_7B | BoN (top-$p$) | -2.58 | -4.78 |
| & UltraRM | AISP | -2.55 | -4.74 |
| Gemma3_4B | BoN (top-$p$) | -4.92 | -5.24 |
| & UltraRM | AISP | -4.41 | -5.24 |
| Llama3_8B | BoN (top-$p$) | -6.88 | -5.42 |
| & Eurus | AISP | -6.95 | -5.13 |
| Vicuna_7B | BoN (top-$p$) | -4.25 | -4.87 |
| & Eurus | AISP | -4.16 | -4.87 |
| Gemma3_4B | BoN (top-$p$) | -7.36 | -5.35 |
| & Eurus | AISP | -7.03 | -5.38 |

# E  ADDITIONAL RELATED WORK

Chakraborty et al. (2024) have presented transfer-$Q^*$ that estimates token-level value function through the trajectory-based reward function. When the base LLM is not aligned with the given target reward in advance, transfer-$Q^*$ requires the base reward model, and it is difficult to fairly compare AISP with it. Xu et al. (2025) have presented an autoregressive reward model GenARM for the test-alignment of LLMs. Chen et al. (2025) have Personalized Alignment at Decoding-time (PAD), which is a framework to align LLM outputs with diverse personalized preference. These method use the decoding method similar to ARGS, and AISP can be used to optimize their reward functions.

# F  RUNTIME EVALUATION

To investigate the computation cost of AISP in detailed, we evaluate the runtime of AISP on a stand-alone server (A100 VRAM 40GB). For fair comparison, we set the generated token length to fixed 128. First, to confirm that the overhead for weight updating Eq. (14) is trivial as explained

Table 6: Alpaca-Eval 2.0

|  | Length controlled winrate | Win rate | Standard error |
|---|---|---|---|
| AISP | 5.64 | 2.86 | 0.59 |
| BoN | 3.95 | 2.24 | 0.52 |

Table 7: GSM8K (8 shot)

|  | Acc |
|---|---|
| AISP | 67.5 |
| BoN | 66.0 |

Table 8: HumanEval

|  | Pass@1 |
|---|---|
| AISP | 41.4 |
| BoN | 34.1 |

Table 9: TruthfulQA

|  | BLEU acc | ROUGE1 acc |
|---|---|---|
| AISP | 0.426 | 0.479 |
| BoN | 0.424 | 0.458 |

Section 3.6, we evaluate the runtimes for one iteration of AISP ($n = 32$) and BoN ($N = 32$) on 10 prompts of SHP with llama3_8B and UltraRM. We observed that AISP and BoN take 7.75 s and 7.68 s for one iteration, respectively. Therefore, the overhead for weight updating at each iteration is about 1%.

Next, we evalauted the runtime for Batched AISP $(b, n) = (4, 8)$ and BoN ($N = 32$) on 100 prompts of SHP with llama3_8B and UltraRM. We observed that Batched AISP takes 935.2 s for while BoN takes 683.9 s. The overhead reaches 36.8 % because the runtime of Batched AISP for one mini-batch is determined by the largest prompts in the mini-batch. After sorting the prompts in terms of the token length, the runtime of Batched AISP becomes 738.4 s, and thus, the overhead is just 8 %.

## G  ADDITIONAL TASKS

To investigate the effectiveness of AISP in various tasks, we compare AISP with BoN on Alpaca-Eval (Li et al., 2023), GSM8K (Cobbe et al., 2021), HumanEval (Chen et al., 2021), and Truth-fulQA (Lin et al., 2022). For the last two tasks, we use the lm-evaluation-harness codes (Gao et al., 2024). We limit token length of responses to 128. The hyper-parameters are the same with the evaluation on SHP with Llama3_8B and UltraRM/Eurus: we set $n = 32$, $\kappa = 32$, and $N = 1024$, and the LLM is llama3_8B. We used UltraRM except for GSM8K and used Eurus for GSM8K because the original paper of Eurus (Yuan et al., 2024) have shown effectiveness on GSM8K. Tables 6-9 show that AISP is effective on these tasks.

## H  LLM USAGE

In addition to using LLMs for experiments, we utilized LLMs to correct grammatical errors and to rephrase some sentences to improve the naturalness of English expressions of some parts of this paper. This procedure was performed by giving the LLMs just partial sentences consisting of few words. We did not perform text generation for larger texts, such as paragraphs exceeding several lines.

