# OpenReview forum: "Test-Time Alignment of LLMs via Sampling-Based Optimal Control in pre-logit space"
_ICLR.cc/2026/Conference — Submitted to ICLR 2026_

### Official Review · Reviewer_GUD6 · 2025-10-29

**Soundness:** 2
**Presentation:** 1
**Contribution:** 2
**Rating:** 4
**Confidence:** 3

**Summary:**

This paper proposes a new method for aligning large language models at test time without fine-tuning. It works by adding Gaussian perturbations to the model’s pre-logit outputs and using importance sampling to adaptively update the perturbation mean for maximizing expected rewards. The method achieves higher reward efficiency and better alignment than existing test-time approaches (for example, BestOfN).

**Strengths:**

1. The method seems novel, which uses the optimal formulation for the test-time alignment problem.

**Weaknesses:**

1. Lack of important literature review and baseline comparison on more recent test-time alignment papers, such as GenARM [1], PAD [2], which performs better than ARGS that is discussed in this paper. Moreover, GemARM also tries to maximize the value function, so it can be worthwhile to compare with it conceptually or mathematically.
2. The motivation of using the optimal control formulation is not clear. KL-constrained reinforcement learning is a commonly used framework, and it is not clear what the benefits are using the optimal control formulation discussed in this paper.
3. The method assumes that the pre-logits follow a Gaussian distribution. This does not seem to hold in practice.

[1] GenARM: Reward guided generation with autoregressive reward model for test-time alignment. In International Conference on Learning Representations, 2025.
[2] PAD: Personalized alignment at decoding-time. In International Conference on Learning Representations, 2025.

**Questions:**

What is the main benefits of using the optimal control formulation instead of the commonly used RL formulation?

---

> ### Author Response · Authors · 2025-11-20
> **Response to GUD6**
>
> We would like to thank the reviewer for the valuable comments and suggestions that helped us clarify and refine the paper!
> > Lack of important literature review and baseline comparison on more recent test-time alignment papers, such as GenARM [1], PAD [2], which performs better than ARGS that is discussed in this paper. Moreover, GemARM also tries to maximize the value function, so it can be worthwhile to compare with it conceptually or mathematically.
> [1] GenARM: Reward guided generation with autoregressive reward model for test-time alignment. In InternationalConference on Learning Representations, 2025.
> [2] PAD: Personalized alignment at decoding-time. In InternationalConference on Learning Representations, 2025.
>
> Thank you for the information about important literature, and we will add the discussions in related work!
> Though they are important related studies, our paper has the different scope from [1] and [2], and thus, they are not direct baselines as follows:
> **Our paper focuses on the decoding algorithm to maximize given reward models, whereas [1] and [2] focuses on designing reward models.**
>  GenARM [1] proposes a training method for token-wise auto-regressive reward models, and the main contribution of PAD [2] is to design the personalized reward models. In fact, **the decoding algorithms for GenARM (eq.(7) in [1]) and PAD (eq.(13) in [2]) seem almost the same as ARGS**: They decode the responses by using the product of the  probability distribution of an LLM and the reward, or the sum of their logarithms. It might be interesting to apply AISP to the reward functions designed by these methods.
>
> > The method assumes that the pre-logits follow a Gaussian distribution. This does not seem to hold in practice.
>
> Gaussian assumption for pre-logits (a.k.a. embeddings) is a well-used heuristic [Zhu et al. 2025, a, b]. For example, Zhu et al. (2025) inject Gaussian noise into input embeddings for Bayesian optimization. The study of [a] models embeddings of BERT by using Gaussian distributions. The study of [b] uses Mahalanobis distance between embeddings, which can be derived from the Gaussian distribution assumption.
>
> Nevertheless, the Gaussian assumption remains useful for our objective even if it does not perfectly match the actual distribution. This is because our objective is to optimize given rewards rather than to fit the data. Theorem 3.2 explains that the Gaussian assumption enables importance sampling-based optimization algorithms, and the other distribution makes it difficult (Section 3.4 lines 223-229). Our experiments confirm its effectiveness.
>
> [a] B. Li, et al., "How is BERT surprised? Layerwise detection of linguistic anomalies", ACL2021
> [b] A. Podolskiy, et al. "Revisiting Mahalanobis Distance for Transformer-Based Out-of-Domain Detection," AAAI2021
>
> > What is the main benefits of using the optimal control formulation instead of the commonly used RL formulation?
>
> First, we would like to emphasize that the optimal control and RL are closely related rather than mutually exclusive methodologies. In fact, the public reward functions used in our experiments were obtained beforehand through an RL-based approach. In this sense, our approach is best viewed as a mixture of optimal control and RL: at test time, we repeatedly optimize the trajectory with an explicit reward function through the optimal control.
>
> Compared to RL, the optimal control tends to prioritize theoretical properties such as convergence and stability. In this sense, AISP solves test-time alignment with theoretical guarantees of model predictive control (MPC) established in Theorem 3.1 and 3.2. AISP converges to the optimal trajectory for a sufficiently large number of samples. **Importantly, even within the RL context, we could not find a response optimization method for the given objective function similar to AISP.**

---

> > ### Comment · Reviewer_GUD6 · 2025-11-25
> >
> > Thank you for the clarification. However, I still have the following concerns.
> >
> > 1. The Gaussian assumption: it seems that this is used because it allows to derive closed-form formula, but might not hold in practice. I am not super sure if this makes sense in practice.
> > 2. The algorithm seems very costly during inference, which is almost the same as best-of-n (which is known to be super slow). [1][2] is much faster, although the authors claim that they are not directly comparable because this paper focus on optimizing reward given existing reward functions.
> >
> > [1] GenARM: Reward guided generation with autoregressive reward model for test-time alignment. In InternationalConference on Learning Representations, 2025.
> > [2] PAD: Personalized alignment at decoding-time. In InternationalConference on Learning Representations, 2025.

---

> > > ### Author Response · Authors · 2025-11-26
> > > **Thank you for the feedback!**
> > >
> > > Thank you for the reaction to our reply! We are glad that our response can address your concern about the comparison between optimal control with RL. We would like to respond to your additional comments.
> > >
> > > > The Gaussian assumption: it seems that this is used because it allows to derive closed-form formula, but might not hold in practice. I am not super sure if this makes sense in practice.
> > >
> > > Our objective is to obtain the optimal response, and the Gaussian distribution is employed for this search. The advantage of closing this distribution to the practical distributions lies in the efficiency of the search. Thus, the more sophisticated distribution assumptions might potentially improve search efficiency. However, the trade-off is that the optimization algorithm becomes more complex, as explained. While the optimization efficiency may not be optimal, the effectiveness of AISP has been demonstrated across broader tasks, as shown in the additional dataset experiments of general response.
> > >
> > > > The algorithm seems very costly during inference, which is almost the same as best-of-n (which is known to be super slow).
> > >
> > > We respectfully note that best-of-n (BoN) is slower than [1] and [2] only in a limited situation. When parallel computation is possible, the overhead is evaluation by the reward model, which is a shared step in [1] and [2]. Section 6.2 of [1] and Table C2 of [2] claim that GenARM and PAD are much faster compared to BoN performing N computations **sequentially**. However, in practice, BoN is computed **as parallelly as possible**.
> > >
> > > On the other hand, ARGS, which is the similar optimization method employed by [1] and [2], is also not very fast because it evaluates rewards for each token generation.  The original paper of ARGS claims that the computation cost is $O(k m^2)$ for $m$ tokens with top-$k$. This is the same as BoN when $k=N$.
> > > In our experiments, the computation time of ARGS with UltraRM for 100 prompts of SHP becomes about 5300 s when $k=7$, whereas BoN ($N=32$) for 100 prompts takes 683.9 s.
> > >
> > > Since Batched AISP operates at nearly the same speed as BoN, we believe that AISP is not super slow compared to the optimization method used in [1] and [2]. In addition, since AISP is the optimization method for given rewards, our method can optimize the reward designed by [1] and [2].

---

> ### Author Response · Authors · 2025-12-02
> **Comparison with GenARM [1]**
>
> We evaluate GenARM [1] on our experimental setting, and our quick experiment demonstrates that **Batched AISP can outperform it with small overhead:** GenARM achieves -5.43 reward values on HHRLHF and  takes 8.59 s/prompt.
> On the other hand, Batched AISP (b,n)=(4,8) achieves -5.28 reward values and takes 9.96 s/prompt. **GenARM requires training time besides the above inference time.**
> To emphasize once more, the main contributions of GenARM [1] and PAD [2] lie in the design of the reward model, and its scope differs from that of AISP, and it is difficult to use PAD [2] for the improvement of the given reward, like AISP, because PAD [2] is a method to align LLM outputs with diverse personalized preferences.

---

### Official Review · Reviewer_Wd2o · 2025-10-30

**Soundness:** 4
**Presentation:** 3
**Contribution:** 3
**Rating:** 6
**Confidence:** 4

**Summary:**

This paper proposes a training-free method called Adaptive Importance Sampling on Pre-logits (AISP), a test-time alignment technique that leverages stochastic model predictive control. By injecting Gaussian noise into the pre-logits and optimizing the perturbation mean to maximize the expected reward, AISP eliminates the need for data collection or additional computation during training. Experimental results show that AISP outperforms Best-of-N and other reward-based test-time alignment methods.

**Strengths:**

- AISP eliminates the need for training and data collection in test-time alignment.
- The integration of adaptive importance sampling with model predictive path integral (MPPI) control is novel and well-motivated.
- The analysis of modeling pre-logits $z$ as Gaussian distributions and its connection to Best-of-N (BoN) is insightful.

**Weaknesses:**

- AISP introduces numerous hyperparameters, including the standard deviation $\sigma$, the softmax temperature $\lambda$, MPPI coefficient $\alpha$, number of iterations $k$, and window size $\tau$. This complexity limits the practicality of AISP. Moreover, the paper does not sufficiently analyze the sensitivity of these hyperparameters across different tasks and models, or their interactions with standard generation parameters such as temperature, top-$p$, and top-$k$ sampling.
- Although AISP shows promise, its improvements over existing methods (as reported in Table 1) appear relatively small improvement, particularly in terms of diversity and coherence.
## References
[1] Scaling Laws for Reward Model Overoptimization. ICML 2023.

[2] BOND: Aligning LLMs with Best-of-N Distillation. ICLR 2025.

**Questions:**

- A deeper discussion of the hyperparameters $\alpha$ and $\tau$ is needed. What specific roles do they play? Why not set $\tau$ equal to the full sequence length $T$? Is there a configuration of hyperparameters that remains robust across different models and settings?
- Since BoN demonstrates strong robustness under large sampling budgets in mitigating reward over-optimization [1, 2], it is important to examine whether AISP maintains better robustness in the Reward-KL regularization trade-off as the sampling budget
$𝑘$ increases. Does AISP still outperform BoN under large budgets (e.g., $k=\{128,256\}$)? Between scaling the number of responses per iteration and the number of iterations, which factor is more vulnerable to reward over-optimization?
- Although the authors claim that AISP and BoN have comparable sequential and parallel computational costs, it's still necessary to include empirical measurements of inference time under identical sampling budgets. As AISP involves additional Gaussian sampling and iterative importance weight updates, quantifying this overhead is essential.
- While not really necessary, it would be informative to compare AISP with standard RLHF training methods such as PPO and DPO.

---

> ### Author Response · Authors · 2025-11-20
> **Response to Wd2o 1/2**
>
> Thank you for careful reading and several comments based on deep understanding of our paper!
> If you have new concerns, we would be glad to hear them by the deadline for reviewer-author discussion.
> For clarity, the following responses have been reordered to distinguish between the concerns bout hyperparameters and about performance.
> # About hyperparameters
> > AISP introduces numerous hyperparameters, including the standard deviation, the softmax temperature, MPPI coefficient, number of iterations, and window size.
>
> First, we would like to clarify that temperature, top-p, and $\tau$ do not require tuning in AISP. AISP does not use temperature and top-p because AISP employees greedy sampling for softmax output (Line 10 in Algorithm 1 of Appendix B). If the window size means $\tau$, $\tau$ is set to max new tokens in practice. In addition, since Fig. 3 shows the reward monotonically increases according to the number of iterations, the number of iterations can be determined by the computation budget.
>
> **Thus, the main tuning-required hyper-parameters of AISP are the following three: $\alpha$, $\lambda$, and $\sigma$.**
>
> > A deeper discussion of the hyperparameters $\alpha$ and $\tau$ is needed. What specific roles do they play? Why not set equal to the full sequence length?
>
> As you say, $\tau$ is set to full sequence length.
> The reason for introducing $\tau$ is that, theoretically, the response length $T$ cannot be predetermined until EOS is generated.
> However, in practice, LLMs have max-new-tokens parameter, and we set $\tau$ to it. $\alpha$ is similar effect to $\lambda$. When $\alpha \rightarrow 1$, AISP allows the resulting distribution to deviate further from the base distribution: The KL divergence of AISP $\alpha=0.9999$ is 90.6 while that of AISP $\alpha=0.99$ is 18.9 in Table 3.
> This effect enables a well-balanced KL divergence without reducing $\lambda$ excessively. If we do not use $\alpha$, reducing $\lambda$ causes AISP to converge asymptotically to BoN as demonstrated in Theorem 3.3. Fig. 7 (c) shows that $\alpha$ should be larger than 0.99, and the condition of $\alpha=0.999$ is effective in most cases in Table 4 in Appendix.
> For the other cases (Lllama&UltraRM (L&U) on SHP, Vicuna&UltraRM (V&U) on SHP, Vicuna&UltraRM (V&U) on HHRLHF, and Vicuna&Eurus (V&E) on HHRLHF), we additionally evaluate AISP with the fixed $\alpha=0.999$ and observed that AISP still outperforms BoN:
>
> **Table E Evaluation of the setting $\alpha=0.999$**
> |Method|L&U on SHP|V&U on SHP|V&U on HHRLHF|V&E on HHRLHF|
> |-|-|-|-|-|
> |Bon|-2.55| -1.80|-4.97|-4.97|
> |AISP|-1.30| -1.76|-4.77|-4.91|
>
> Due to the limited rebuttal period, the above experiment was conducted on 100 prompts.
> **Therefore, $\alpha=0.999$ is effective on all conditions in our experiments, and
> we can reduce the hyperparameter $\alpha$ by fixing it to 0.999.**
>
> >  Is there a configuration of hyperparameters that remains robust across different models and settings?
>
> Hyper-parameters are tend to sensitive to reward models but robust against datasets and LLMs. In fact, AISP achieves good performance on additional datasets in the above general response although we do not tune hyper-parameters: setup is the same as that of SHP dataset. Sensitivity to reward models is caused by various range of outputs of reward models: Eurus outputs thousands of value ranges, whereas UltraRM outputs only tens.

---

> ### Author Response · Authors · 2025-11-20
> **Response to Wd2o 2/2**
>
> # About performance
> > Although AISP shows promise, its improvements over existing methods (as reported in Table 1) appear relatively small improvement, particularly in terms of diversity and coherence
>
> To clarify the effectiveness of AISP, we evaluate the additional datasets (Alpaca-Eval, GSM8K, HumanEval, and TruthfulQA) as described in the general responses. AISP outperforms BoN on all tasks, including code and math tasks.
>
> > Since BoN demonstrates strong robustness under large sampling budgets in mitigating reward over-optimization [1, 2], it is important to examine whether AISP maintains better robustness in the Reward-KLregularization trade-off as the sampling budget
> increases. Does AISP still outperform BoN under large budgets(e.g.,
> )?
>
> Thank you for the comments based on your expertize!
> As you say, the advantage of BoN is effectiveness on large sampling budgets, and thus, we compare BoN with large $N=1024$ to AISP in Table 1. Following your comments, we evaluate $k=128 (N=4096)$ for further large sampling budget. **AISP achieves higher reward -0.46 than that of BoN -1.46, and their win rate becomes (BoN, draw, AISP)=(38, 6, 56) on 100 prompts of SHP. On 100 prompts of HHRLHF, rewards become BoN: -5.23 and AISP: -5.02, and win rateis (32: 18: 50).** These results show a similar trend to the improvement in AISP performance with respect to the number of samples in Fig. 3. Note that we increases $\lambda$ to 1.0 because increase of $\kappa$ moves the distribution far from the base model (reward on SHP of $\lambda=0.3$: -0.9518).
>
> Note that we randomly selected 100 prompts for each dataset in this experiments due to computation costs. And thus, we cannot directly compare the above results with Table 1.
>
> > Between scaling the number of responses per iteration and the number of iterations, which factor is more vulnerable to reward over-optimization?
>
> This is a difficult problem. Fig. 4 suggests that large iterations reduce the variance in results, while increasing the number of responses per iteration achieves higher rewards as following:
> Since batch size is equal to the number of iteration $b=\kappa$, Fig 4 evaluates the importance of the responses per iteration $n$ with the number of iterations $\kappa$. AISP1 $(\kappa, n)$=(16, 8) is larger averaged reward values than AISP3 $(32, 4)$ though its error-bar is larger than that of AISP3. Large iterations with moderate sample size might be a stable and effective choice.
>
> > it's still necessary to include empirical measurements of inference time under identical sampling budgets. As AISP involves additional Gaussian sampling and iterative importance weight updates, quantifying this overhead is essential.
>
>  First, runtimes for one iteration of AISP ($n=32$) and BoN ($N=32$) are 7.75 s and 7.68, respectively. Thus, **the overhead for weight update at each iteration is about 1 %**, which is averaged results over 10 prompts of SHP with llama3_8B and UltraRM.
>
> Next, Batched AISP $(b,n)=(4,8)$ takes 935.2 s for 100 prompts while BoN ($N=32$) takes 683.9 s (36.8 % overhead).
> **After sorting the prompts in terms of the token length, the runtime of Batched AISP becomes 738.4 s  and thus, the overhead is just 8 %.**
> Thank you for your astute comments! They provided interesting and important results, and we will add discussions! The detailed setup and results can be found in general response.
>
> > While not really necessary, it would be informative to compare AISP with standard RLHF training methods such as PPO and DPO.
>
> Thank you for your useful comment. Though we were unable to conduct experiments due to priorities of other your important comments, we expect the following results to be obtained: Since baseline methods, RE-Control and ARGS, outperform LoRA-based PPO and DPO in (Khanov et al. 2024, Kong et al. 2024), AISP can outperform them.

---

### Official Review · Reviewer_w4ta · 2025-11-02

**Soundness:** 2
**Presentation:** 2
**Contribution:** 2
**Rating:** 4
**Confidence:** 4

**Summary:**

The paper proposes AISP, a training-free, test-time alignment method that injects Gaussian perturbations into the pre-logits over a fixed control window and updates the perturbation mean by adaptive importance sampling. The objective is cast via a free-energy bound, yielding an optimal pre-logit distribution proportional to $exp(r/\lambda)p(V)$, since this is intractable, the mean is iteratively estimated with weighted samples. The authors argue the Gaussian assumption is consistent with softmax classifiers, show that AISP reduces to best-of-N (BoN) as as $\lambda$→0, and report higher reward and GPT-4 win-rates than BoN, ARGS, and RE-Control on SHP/HH-RLHF.

**Strengths:**

1. This paper maps decoding-time reward maximization to sampling-based optimal control in pre-logit space; derivation via a free-energy lower bound is standard but cleanly presented.

2. The method is training-free and lightweight; also the adaptive importance sampling loop is easy to implement.

3. Empirical results show consistent reward and win-rate improvements over BoN with the same total samples.

**Weaknesses:**

1. The control-theoretic view and MPPI-style derivation are known; the main step is moving importance sampling to pre-logit trajectories with a Gaussian prior. The reduction to BoN for λ→0 underscores AISP as a structured BoN generalization to me rather than a new paradigm.

2. Tasks are preference datasets (SHP, HH-RLHF) with reward-model scoring; diversity/coherence sometimes degrade, and win-rate uses small paired samples. No tests on reasoning/code/math where long-horizon dynamics might stress the method.

3. AISP requires sequential updates; the paper does not report wall-clock vs. BoN under the same accelerator budget, nor scaling under constrained interleave with other requests.

4. The Gaussian model for pre-logits is only heuristically connected to softmax; no quantitative validation of this assumption.

**Questions:**

1. Can you quantify the Gaussian pre-logit assumption (per-token goodness-of-fit, across layers/models)? Any evidence that non-Gaussian priors would materially help/hurt AISP?

2. Please report wall-clock and throughput (tokens/sec) vs. BoN for matched total samples under realistic GPU budgets?

3. For RE-Control, what data/compute were used to train the value function, and how does AISP compare at equal wall-clock including that one-time cost?

---

> ### Author Response · Authors · 2025-11-20
> **Response to Reviewer w4ta 1/2**
>
> We appreciate your thoughtful comments, and we are happy to improve our paper based on your comments!
> > The control-theoretic view and MPPI-style derivation are known; the main step is moving importance sampling to pre-logit trajectories with a Gaussian prior. The reduction to BoN for λ→0 underscores AISP as a structured BoN generalization to me rather than a new paradigm.
>
> As you say, MPPI has already been established method. Our contribution is (i) modifying it by integrating adaptive importance sampling and (ii) its formulation of pre-logit optimization for test-time alignment, and (ii) confirmation of its effectiveness through experiments. The first two points are acknowledged by Reviewer W2d, and the last point is acknowledged by Reviewer utag. Additionally, we believe the connection between AISP and BoN is an important result because this reveals that there is a more generalized extension of well-used BoN, and it achieves better performance.
>
> > No tests on reasoning/code/math where long-horizon dynamics might stress the method.
>
> Following your comments, we evaluate Alpaca-Eval, GSM8K (math), TruthfuQA, and HumanEval (code). On all tasks, AISP outperforms BoN. This indicates that AISP is effective in general tasks.
> We are grateful for your comments, which guided us to strengthen the paper!
>
> **Table A Alpaca-Eval 2.0**
> |Method| length controlled winrate |win_rate|standard_error|
> |-|-|-|-|
> |AISP|5.64| 2.86|0.59|
> |BoN|3.95|2.24|0.52|
>
> **Table B GSM8K with 8 shot**
> |Method| Acc|
> |-|-|
> |AISP| 67.5|
> |BoN|66.0|
>
> **Table C HumanEval**
> |Method| pass@1|
> |-|-|
> |AISP| 41.4|
> |BoN|34.1|
>
> **Table D TruthfulQA**
> |Method|BLEU acc|ROUGE1 acc|
> |-|-|-|
> |AISP|0.426| 0.479|
> |BoN|0.424|0.458|
>
>
>
> The detailed setup is shown in general response.
>
> > AISP requires sequential updates; the paper does not report wall-clock vs. BoN under the same accelerator budget, nor scaling under constrained interleave with other requests.
>
> Though AISP needs sequential updates, we can utilize mini-batch computation (Section 3.6) if we need to reduce the runtime. Thanks to the sample-efficiency, Batched AISP outperforms BoN (Section 5.3). We agree on the importance of the wall-clock evaluation and observed that the overhead of Batched AISP is just 8% as the next response!
>
> > Please report wall-clock and throughput (tokens/sec) vs. BoN for matched total samples under realistic GPU budgets?
>
> The throughput of Batched AISP is 17.3 tokens/sec, while that of BoN is 18.7 tokens/sec though they include the cost for processing the prompts (738.4 s for 128 tokens per/prompt $\times$ 100 prompts). The 8 % overhead is caused by the fact that the largest input prompt in the mini-batch determines the runtime of AISP. We are glad that we have obtained results that deepen our understanding of AISP! The detailed results and setup can be found in general response.

---

> ### Author Response · Authors · 2025-11-20
> **Response to Reviewer w4ta 2/2**
>
> > The Gaussian model for pre-logits is only heuristically connected to softmax; no quantitative validation of this assumption.
> > Can you quantify the Gaussian pre-logit assumption (per-token goodness-of-fit, across layers/models)?
>
> It is difficult to evaluate the assumption due to the difficulty of sampling and high dimensionality. First, unbiased sampling $\mathbf{z}_t$ from $p(\mathbf{z}_t|y_t)$ is difficult because the exact dataset trained by LLMs is unknown and too large data size. Second, $\mathbf{z}_t$ has too high dimension to use the multi-variable normality test, such as Henze-Zirkler's test.
>
> Nevertheless, the Gaussian assumption for pre-logits (a.k.a. embeddings) is a well-used heuristic [Zhu et al. 2025, a, b]. For example, Zhu et al. (2025) inject Gaussian noise into input embeddings to use Bayesian optimization. The study of [a] models embeddings of BERT by using Gaussian distributions, and the study of [b] uses Mahalanobis distance between embeddings, which can be derived from the Gaussian distribution assumption.
>
> We would like to emphasize that the Gaussian assumption does not need to match the actual distribution perfectly as long as AISP optimizes the response because our objective is not to fit the data. Our experiments demonstrate that the Gaussian assumption is useful to optimize the response.
>
> [a] B. Li, et al., "How is BERT surprised? Layerwise detection of linguistic anomalies", ACL2021
> [b] A. Podolskiy, et al. "Revisiting Mahalanobis Distance for Transformer-Based Out-of-Domain Detection," AAAI2021
>
> >  Any evidence that non-Gaussian priors would materially help/hurt AISP?
>
> Theorem 3.2 explains that the Gaussian assumption enables importance sampling-based optimization algorithms, and Lines 223-229 (Section 3.4) explain that the other distribution makes it difficult. Specifically, the Gaussian distribution helps AISP because its KL divergence from the optimal distribution $q^*$ (Eq.(10)) is minimized by $\mathbb{E}[\mathbf{v}_t]$, which is approximated by importance sampling. In the context of MPPI literature, Power & Berenson (2023) propose a more flexible distribution, but it requires the normalizing flow, which incurs additional training costs.
>
> If you have other ideas about candidate distributions, it would be our pleasure to consider their feasibility during the discussion!
>
> > For RE-Control, what data/compute were used to train the value function, and how does AISP compare at equal wall-clock including that one-time cost?
>
> Datasets of RE-Control are composed by responses and pre-logits $\mathbf{z_t}$ of all tokens for each training dataset of SHP and HHRLHF. For example, SHP has 349,000 training prompts and the generated dataset requires 320 GB storage cost. **All datasets of RE-Control in our experiments achieve 1.3TB**, which is large storage cost.
> **For equal wall-clock, we observed that AISP still outperforms RE-Control as follows:**
> RE-control requires 4.5 hour/dataset training time and 12.24 s/prompt inference-time. Since AISP takes 7.75 s/iteration for one iteration, the inference time becomes almost the same when $\kappa$=2. For $\kappa$=2, the reward of AISP achieves -4.63, which is larger than RE-Control -9.28 on SHP with llama3_8B and UltraRM13B. Similarly, the reward of AISP becomes -5.24, which is larger than RE-Control  -5.53 on HHRLHF. We are happy that the new results strength our paper!

---

### Official Review · Reviewer_uatg · 2025-11-03

**Soundness:** 2
**Presentation:** 3
**Contribution:** 2
**Rating:** 4
**Confidence:** 3

**Summary:**

In this paper, the authors propose an inference-time alignment framework, AISP (Adaptive Importance Sampling on Pre-logits). The authors frame this alignment task as a sampling-based optimal control problem. The core idea is to apply a stochastic Gaussian perturbation to the pre-logits, i.e., the penultimate layer outputs at each decoding step. AISP then uses adaptive importance sampling to iteratively update the mean of this perturbation, effectively creating a control signal that explores the generation space and guides the model toward high-reward sequences. Experimental evaluations on standard benchmarks on HH and SHP show that AISP outperforms competitive baselines such as  BoN, ARGS, and RE-Control.

**Strengths:**

1. The paper is generally well written. The proposed AISP approach operates at inference time and therefore does not require training value functions, unlike RE-Control.

2. The authors provide detailed hyperparameter ablations, a KL-divergence analysis, and a thorough comparison of batched AISP with BoN.

3. The empirical evaluation is comprehensive, covering multiple base LLMs and reward models.

**Weaknesses:**

1. The performance improvement on HH-RLHF appears incremental, and in many cases, BoN outperforms AISP. With only two datasets, it is difficult to fully assess AISP’s empirical effectiveness. I recommend evaluating on additional datasets to more clearly demonstrate the gains.

2. [Minor] While the paper includes strong baselines, adding comparisons with the controlled decoding literature [1, 2] would further strengthen the experimental section.

[1] Mudgal, S., Lee, J., Ganapathy, H., Li, Y., Wang, T., Huang, Y., Chen, Z., Cheng, H.T., Collins, M., Strohman, T. and Chen, J., 2023. Controlled decoding from language models. arXiv preprint arXiv:2310.17022.

[2] Chakraborty, S., Ghosal, S.S., Yin, M., Manocha, D., Wang, M., Bedi, A.S. and Huang, F., 2024. Transfer q-star: Principled decoding for llm alignment. Advances in Neural Information Processing Systems, 37, pp.101725-101761.

**Questions:**

Please see weakness 1.

---

> ### Author Response · Authors · 2025-11-20
> **Response to Reviewer utag**
>
> We appreciate your helpful comments! By your suggestion, we obtain more convincing experimental results as follows:
>
> > W1 With only two datasets, it is difficult to fully assess AISP’s empirical effectiveness. I recommend evaluating on additional datasets to more clearly demonstrate the gains.
>
> As explained in general response, we compare AISP with BoN on Alpaca-Eval, HumanEval, TruthfulQA, and GSM8K. AISP outperforms BoN on the all tasks. We limit token length of responses to 128 and use llama3_8B and UltraRM-13B, and the detailed setup can be found in general response. We are thankful for the the helpful comments, which led us to make improvements!
>
> **Table A Alpaca-Eval 2.0**
> |Method| length controlled winrate |win_rate|standard_error|
> |-|-|-|-|
> |AISP|5.64| 2.86|0.59|
> |BoN|3.95|2.24|0.52|
>
> **Table B GSM8K with 8 shot**
> |Method| Acc|
> |-|-|
> |AISP| 67.5|
> |BoN|66.0|
>
> **Table C HumanEval**
> |Method| pass@1|
> |-|-|
> |AISP| 41.4|
> |BoN|34.1|
>
> **TableD TruthfulQA**
> |Method|BLEU acc|ROUGE1 acc|
> |-|-|-|
> |AISP|0.426| 0.479|
> |BoN|0.424|0.458|
>
> If you have any additional concerns, we are happy to hear them!
>
> > W2 [Minor] While the paper includes strong baselines, adding comparisons with the controlled decoding literature[1, 2] would further strengthen the experimental section
> [1] Mudgal, S., Lee, J., Ganapathy, H., Li, Y., Wang, T., Huang, Y., Chen, Z., Cheng, H.T., Collins, M., Strohman, T. andChen, J., 2023. Controlled decoding from language models. arXiv preprint arXiv:2310.17022.
> [2] Chakraborty, S., Ghosal, S.S., Yin, M., Manocha, D., Wang, M., Bedi, A.S. and Huang, F., 2024. Transfer q-star:Principled decoding for llm alignment. Advances in Neural Information Processing Systems, 37, pp.101725-101761.
>
> Thank you for the information about important work! We will add the discussion in related work. Nevertheless, [1] and [2] might not be direct comparison for our work because it requires the training of value function similar to RE-Control. Further, (Kong et al., 2024) reported that RE-Control outperforms [1]. [2] requires two reward functions, baseline rewards and target rewards, in an indirect transfer setting that corresponds to our experiments. Thus, it might be difficult to fairly compare it with AISP under our setting.

---

### Author Response · Authors · 2025-11-20
**General responses**

We are deeply grateful for your careful reading and constructive comments. We are happy that Reviewers Wd2o and GUD6 confirm the novelty of our proposed method, and Reviewers w4ta and Wd2o acknowledged the motivation for the training-free alignment, Reviewer uatg recognized the comprehensiveness of the experiments.
If our response contains any gaps or raises new concerns, could you inform us before the deadline?

Before the response to each comment, this form gives general response to the shared concerns.
# Additional datasets
Sine Reviewers utag, w4ta, and Wd2o raise the concern about limited datasets and significance of empirical results, we compare AISP with BoN on Alpaca-Eval, GSM8K, HumanEval, and TruthfulQA.
**AISP outperforms BoN on the all tasks.** For the last two tasks, we use the lm-evaluation-harness codes.
We limit token length of responses to 128.
The hyper-parameters are the same with the evaluation on SHP with Llama3_8B and UltraRM/Eurus.


**Table A Alpaca-Eval 2.0**
|Method| length controlled winrate |win_rate|standard_error|
|-|-|-|-|
|AISP|5.64| 2.86|0.59|
|BoN|3.95|2.24|0.52|

**Table B GSM8K with 8 shot**
|Method| Acc|
|-|-|
|AISP| 67.5|
|BoN|66.0|

**Table C HumanEval**
|Method| pass@1|
|-|-|
|AISP| 41.4|
|BoN|34.1|

**TableD TruthfulQA**
|Method|BLEU acc|ROUGE1 acc|
|-|-|-|
|AISP|0.426| 0.479|
|BoN|0.424|0.458|

- Setup
max new tokens: 128, $n=32$, $\kappa$=32, $N=1024$, Base LLM: llama3_8B, Reward model: UltraRM/Eurus
We used UltraRM except for GSM8K and used Eurus for GSM8K because the original paper of Eurus (Yuan et al., 2024) have shown effectiveness on GSM8K.
# Runtime evaluations
Since Reviewers w4ta and Wd2o point out the runtime evaluation, we evaluate the runtime of AISP on a stand-alone server.
First, runtimes for one iteration of AISP ($n=32$) and BoN ($N=32$) are 7.75 s and 7.68, respectively. Thus, **the overhead for weight updating at each iteration is about 1 %** (averaged results over 10 prompts of SHP with llama3_8B and UltraRM).

Next, we observed that Batched AISP $(b, n)=(4,8)$ takes 935.2 s for 100 prompts while BoN ($N=32$) takes 683.9 s (36.8 % overhead).
This is because the runtime of Batched AISP for one mini-batch is determined by the largest prompts in the mini-batch.
**After sorting the prompts in terms of the token length, the runtime of Batched AISP becomes 738.4 s  and thus, the overhead is just 8 %.**
The constructive review comments provided interesting and valuable results, and we will add results and discussions!

The setup for this experiment is the following: we set the generated token length to fixed 128 for fair comparison.
Additionally, since our stand-alone servers have limited computation resources (A100 VRAM 40GB), we set $N=32$, $\kappa$=7, $n=4$, and $b=8$ unlike Fig. 4.

---

> ### Author Response · Authors · 2025-11-21
> **Revision upload**
>
> We revise our paper on the basis of the reviewers' comments. The main modifications are as follows:
>
> - To clarify that we use greedy sampling after Gaussian injection in AISP, we add argmax in Equation (5)
> - We add Appendix E to discuss the additional previous studies raised by reviewers.
> - We add Appendix F to evaluate the runtime of AISP.
> - We add Appendix G to evaluate AISP on the various tasks.
> - Reference is updated to include additional previous studies and datasets following the above changes.
>
> Thank you for the reviewer’s constructive comments again, and we are happy to improve the quality of our paper.

---

### Author Response · Authors · 2025-12-03

Dear ACs,

We deeply thank ACs and reviewers for carefully conducting the reviewer process and for many constructive comments again!

We would like to summarize the author-reviewer discussion.
Below, we briefly summarize the strengths identified and how we addressed the main concerns raised in the discussion.

---

**Strength**
Reviewers uatg, w4ta, and Wd2o highlighted the training-free nature of AISP. Reviewer w4ta, Wd2o, and GUD6 recognized the novelty of AISP in terms of the formulation mapping reward maximization to the sampling-based optimization.
Additional strengths noted include: broad coverage of base models and reward models (uatg), ease of implementation (w4ta), and the insightful connection between AISP and BoN (Wd2o).

---

**Reviewer uatg**
The main concern of Reviewer uatg is about the small number of datasets, and we add the four benchmarks: Alpaca-Eval, GSM8K, HumanEval, and TruthfulQA. Our experiments consistently demonstrate that AISP outperforms BoN, which is the strongest baseline in our paper.

**Reviewer w4ta**
In response to the request for reasoning/code/math evaluations, we added experiments on GSM8K (math) and HumanEval (code), where AISP consistently outperforms BoN.

We clarify the benefit of Gaussian assumption by using the Theorems and explanations in the main paper: this assumption realizes the easy implementation, and the other sophisticated distribution requires costly training.

Following the suggestion on run-time, we compared Batched AISP, BoN, and RE-Control. We observed that Batched AISP outperforms RE-Control under the same inference-time and incurs only ~8% overhead over BoN, which we consider acceptable given the performance gains.

**Reviewer Wd2o**
The concern about increased hyperparameters is addressed by noting that AISP introduces only one additional hyperparameter compared to BoN in practice. Furthermore, a new result in response shows that one hyper-parameter $\alpha$ can be fixed as $\alpha=0.999$, keeping tuning cost almost the same as BoN.

The suggested experiment with a larger computation budget further  emphasize AISP's advantages.

We also performed run-time analysis and observed an ~8% overhead for Batched AISP due to varying prompt token lengths within mini-batches. This result contributes to a deeper understanding of the characteristics of Batched AISP.

**Reviewer GUD6**
Reviewer GUD6 informed us of the related work: GenARM [1] and PAD [2]. Our additional experiment shows that AISP can outperform GenARM [1], which can be used for the reward maximization among them, in terms of reward values and training-free characteristics. Though [1] and [2] are important works, the scope of our paper is different from theirs: AISP is the decoding algorithm to maximize given reward models, whereas [1] and [2] are designing methods for autoregressive/personalized reward models.

Reviewer GUD6 questioned the merits of employing optimal control, but the additional reply implies that this concern appears to have been addressed by our response.

Regarding the Gaussian assumption, we clarified that it is a common heuristic and that the assumption gap affects only efficiency.  The experiments in our paper demonstrate that the efficiency of AISP is practically effective.

---

We believe that these additional experiments and clarifications address almost all major concerns and hope this overview is helpful for the meta-review.

Best regards,
Authors

---

### Meta-Review · Area_Chair_trGt · 2025-12-29

**Summary:**

This paper proposes AISP, a training-free test-time alignment method that formulates reward maximization as a sampling-based optimal control problem in pre-logit space. Reviewers find the paper technically detailed and empirically thorough, and acknowledge the novelty of framing decoding-time alignment via stochastic control. However, several critical concerns remain unresolved.

(i) The motivation for adopting an optimal control formulation is not sufficiently convincing. While the authors argue for theoretical guarantees and connections to MPPI, reviewers note that the problem can already be naturally formulated under standard KL-regularized RL or inference-time reward maximization frameworks. The paper does not clearly articulate what practical or conceptual advantages the optimal control perspective brings beyond re-deriving a structured generalization of best-of-n sampling.

(ii) The method relies on a strong Gaussian assumption over pre-logits, which reviewers find inadequately justified. The assumption appears primarily chosen for analytical convenience, enabling closed-form importance sampling updates, but there is no empirical validation that pre-logits follow such a distribution in practice. The authors’ responses acknowledge this gap and argue that exact correctness is unnecessary, but this weakens confidence in the soundness and generality of the approach.

(iii) Inference-time efficiency remains a major concern. Despite batched optimizations, AISP requires iterative sampling and reward evaluation, resulting in inference costs comparable to best-of-n, which is already known to be slow. Reviewers point out that recent test-time alignment methods such as GenARM and PAD are substantially faster in practice. While the authors argue these methods are not directly comparable due to differing scopes, the high inference cost of AISP significantly limits its practical appeal, especially given the relatively incremental gains over best-of-n.

Overall, although the paper presents an interesting perspective and solid experimental effort, the unclear motivation for the control formulation, the questionable Gaussian modeling assumption, and the high inference cost collectively undermine the impact and practicality of the contribution. These issues outweigh the demonstrated improvements, and therefore I recommend rejection.

**Reviewer Concerns:**

The rebuttal addressed several empirical concerns by adding evaluations on additional benchmarks, providing more detailed (batched) runtime analyses, and clarifying implementation details, which resolve reviewers’ surface-level questions about experimental scope and reporting.

However, key concerns remain unresolved, including the unclear motivation and added value of the optimal control formulation, the weakly justified Gaussian pre-logit assumption, and the high inference cost that remains comparable to best-of-n and slower than recent alternatives, limiting confidence in the method’s soundness and practicality.

**Reviewer Scores:**

Most of the reviewers still tend to reject this paper with a score of 4,4,4,6.

---

### Decision · Program_Chairs · 2026-01-26

Reject